# Dopamine control of social novelty preference is constrained by an interpeduncular-tegmentum circuit

Susanna Molas [1,2,3] ✉, Timothy G. Freels[1], Rubing Zhao-Shea[1], Timothy Lee [1], Pablo Gimenez-Gomez[1], Melanie Barbini [1], Gilles E. Martin[1] & Andrew R. Tapper [1] ✉

Animals are inherently motivated to explore social novelty cues over familiar ones, resulting in a novelty preference (NP), although the behavioral and circuit bases underlying NP are unclear. Combining calcium and neurotransmitter sensors with fiber photometry and optogenetics in mice, we find that mesolimbic dopamine (DA) neurotransmission is strongly and predominantly activated by social novelty controlling bout length of interaction during NP, a response significantly reduced by familiarity. In contrast, interpeduncular nucleus (IPN) GABAergic neurons that project to the lateral dorsal tegmentum (LDTg) were inhibited by social novelty but activated during terminations with familiar social stimuli. Inhibition of this pathway during NP increased interaction and bout length with familiar social stimuli, while activation reduced interaction and bout length with novel social stimuli via decreasing DA neurotransmission. These data indicate interest towards novel social stimuli is encoded by mesolimbic DA which is dynamically regulated by an IPN→LDTg circuit to control NP.

Social behaviors are dynamic, innate motivational behaviors commonly disrupted in neurodevelopmental and neuropsychiatric disorders[1–3]. Animal species highly react to social novelty cues as compared to familiar ones as an action selection mode that adjusts behavior in ever-changing environments[4,5]. Preference toward novelty over familiarity is critical to evaluate new threats, as well as new opportunities for socializing or mating. However, with repeated exposures, the social novelty response follows habituation, a classic form of behavioral plasticity essential for higher order types of learning[6,7]. The dynamics of behavioral sequences and the neuronal circuitries that coordinate a differential response to social novelty over familiarity are still emerging.

Mesolimbic dopamine (DA) neurotransmission from ventral tegmental area (VTA) DAergic neurons innervating the ventral striatum is engaged by exposure to social cues both in humans[8] and animal models[9–12]. It is generally accepted that mesolimbic DA computes reward prediction error (RPE) signals[13] to update the value of future events and drive learning[14]. Thus, behaviors associated with positive social experiences reflect rewarding events where DA signals predict social cues and learning aspects of sociability[10]. Despite numerous evidence linking DA with social behaviors, the precise activity patterns by which a novel social cue transitions to familiar with repeated exposures, what aspect of social interaction DA encodes, and the real-time DA dynamics during responses of social novelty preference (NP) are unclear. Moreover, our previous work demonstrated the interpeduncular nucleus (IPN) of the midbrain represents a neuroanatomical substrate for familiarity signaling critically involved in responses of NP[15]. As opposed to mesolimbic DA reward-related function, the IPN

[1]Department of Neurobiology, Brudnick Neuropsychiatric Research Institute University of Massachusetts Chan Medical School 364 Plantation St, LRB, Worcester 01605 MA, USA. [2]Institute for Behavioral Genetics, University of Colorado Boulder 1480 30th St, Boulder 80303 CO, USA. [3]Department of Psychology and Neuroscience, University of Colorado Boulder 1905 Colorado Ave, Boulder 80309 CO, USA. ✉e-mail: susanna.molas@colorado.edu; andrew.tapper@umassmed.edu

is emerging as an area that orchestrates avoidance-related behaviors[16], although the dynamics of IPN neurons and their functional output projections are largely unexplored. Here, we identified causal activity patterns of IPN neurons with the termination of familiar social investigations. This activity pattern is conveyed via the laterodorsal tegmental area (LDTg) and constrains social NP responses by direct influence on mesolimbic DAergic signals which encode initial novel social interactions and bout length. Our results provide a detailed analysis of DA activity dynamic patterns underlying NP while also functionally identifying a circuit that may be implicated in numerous disorders associated with impaired social novelty responses.

## Results

### First-novel social interactions increase VTA DAergic neuronal activity patterns and DA neurotransmission which correlates with length of interaction and decreases during familiarity

Mice adopt different sequences of exploratory actions towards novel and familiar stimuli, as well as when given a choice between the two, that reveal the motivational drive of novel investigations[17]. We used the 3-chamber sociability test (Fig. 1a and "Methods"), to elucidate the

activity dynamics of circuits responding to behavioral sequences of novel social investigations. First, we time-locked investigation episodes to VTA DAergic neuronal responses using GCaMP expression in dopamine transporter (DAT) Cre mice combined with fiber photometry recordings synchronized to behavior (Fig. 1b, S1a). Consistent with previous reports[9,10], VTA DAergic neurons significantly increased activity when mice were presented to a novel conspecific both in males (Fig. 1c–e, S1b) and females (Fig. S2a–d). However, VTA DAergic responses with social novelty were lower in females as compared to male mice so sexes were analyzed separately (Two-way ANOVA for mean z-score of the first interaction during days 1–3, main effect of sex: $F_{(1, 7)} = 8.964$, $p = 0.020$ and time: $F_{(1.325, 9.277)} = 7.500$, $p = 0.017$). Remarkably, on day 1 of the test, VTA DAergic neuronal activation was significantly higher when mice initiated a social investigation for the first time with subsequent interactions within the same session eliciting only modest activity (Fig. 1c–e, S1b), indicating that VTA DAergic neuronal activity encodes not only sociability but, importantly, the novelty component of social interactions. Noticeably, DAergic neuronal activity in response to social investigations after the initial interaction shifted to the left, displaying an earlier peak onset of activation

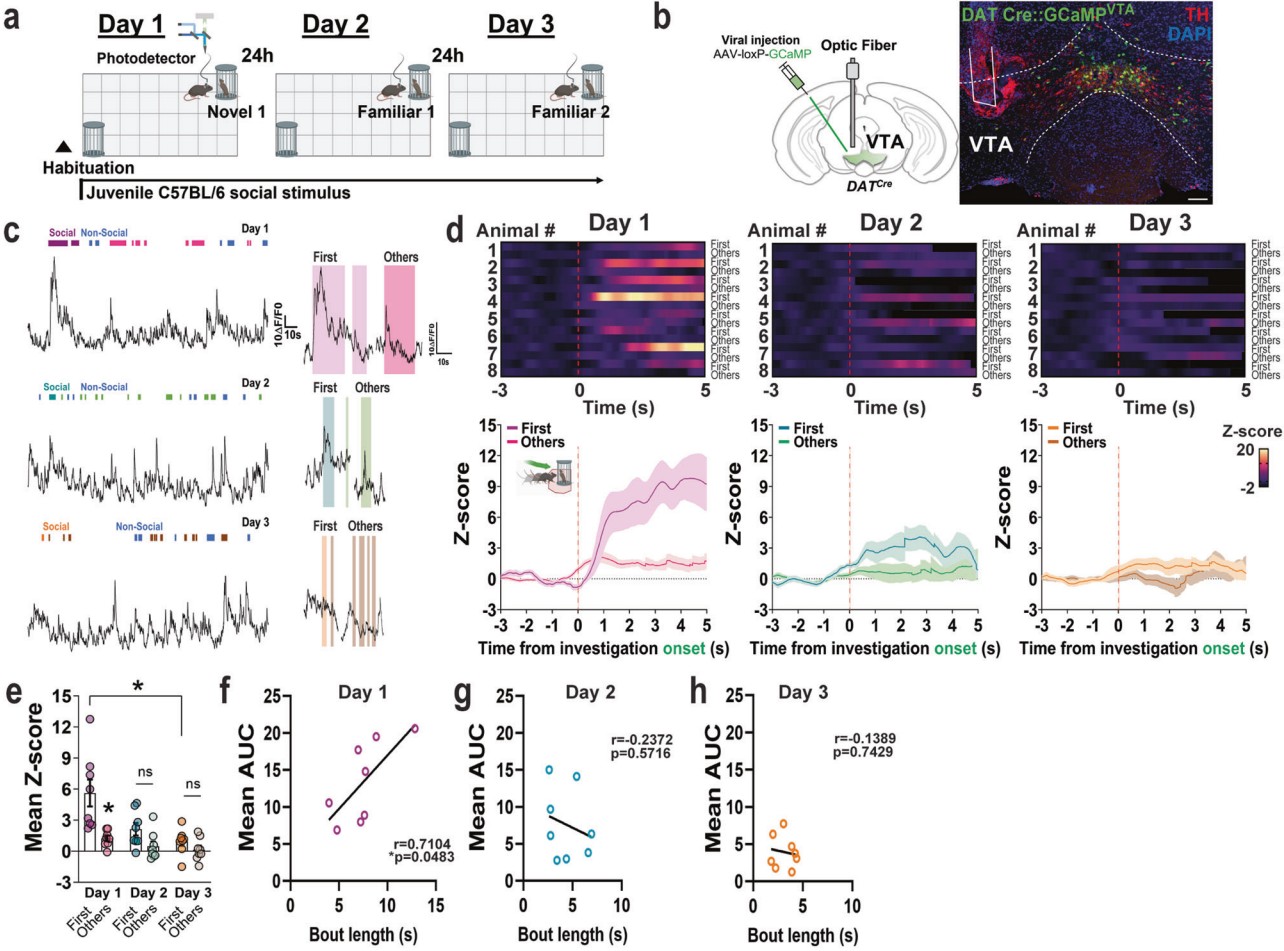

**Fig. 1 | VTA DAergic neuronal activity encodes initial response to social novelty and bout length. a** Schematic, generated using Biorender, of the social behavior paradigm and recording strategy used. **b** Viral injection strategy and representative image of GCaMP6m expression (green signal) in the VTA of DAT[Cre] mice. Neurons are immunolabeled for TH (red) and nuclei are counterstained with DAPI (blue) (n = 8 male mice). Scale bar 100 µm. **c** Representative traces of VTA DAergic activity (dF/F0) during the social paradigm on days 1–3. **d** Heatmap representations and z-scores of time-locked VTA DAergic activity recordings relative to time initiating a social investigation (red line) on Days 1 to 3. The first social investigation is compared to the subsequent ones (n = 8 male mice). **e** Quantification of activity

responses in (d) as mean z-score. Two-way RM ANOVA (time main effect: $F_{(2,14)} = 9.60$, $P = 0.0049$; bout main effect: $F_{(1,7)} = 8.84$, $P = 0.0207$; interaction: $F_{(2,14)} = 8.57$, $P = 0.0073$). Šidák's multiple comparisons *p < 0.05. Correlation between the bout length duration (s) of social investigations and the mean area under the curve (AUC) of VTA GCaMP signals on (**f**) Day 1, Two-tailed Pearson r (r = 0.7104, $R^2$ = 0.5046, p = 0.0483), (**g**) Day 2, Two-tailed Pearson r (r = −0.2372, $R^2$ = 0.0563, p = 0.5716), and (**h**) Day 3, Two-tailed Pearson r (r = −0.1389, $R^2$ = 0.0193, p = 0.7429) of the social paradigm. Data represent mean ± SEM. Source data are provided as a Source Data file.

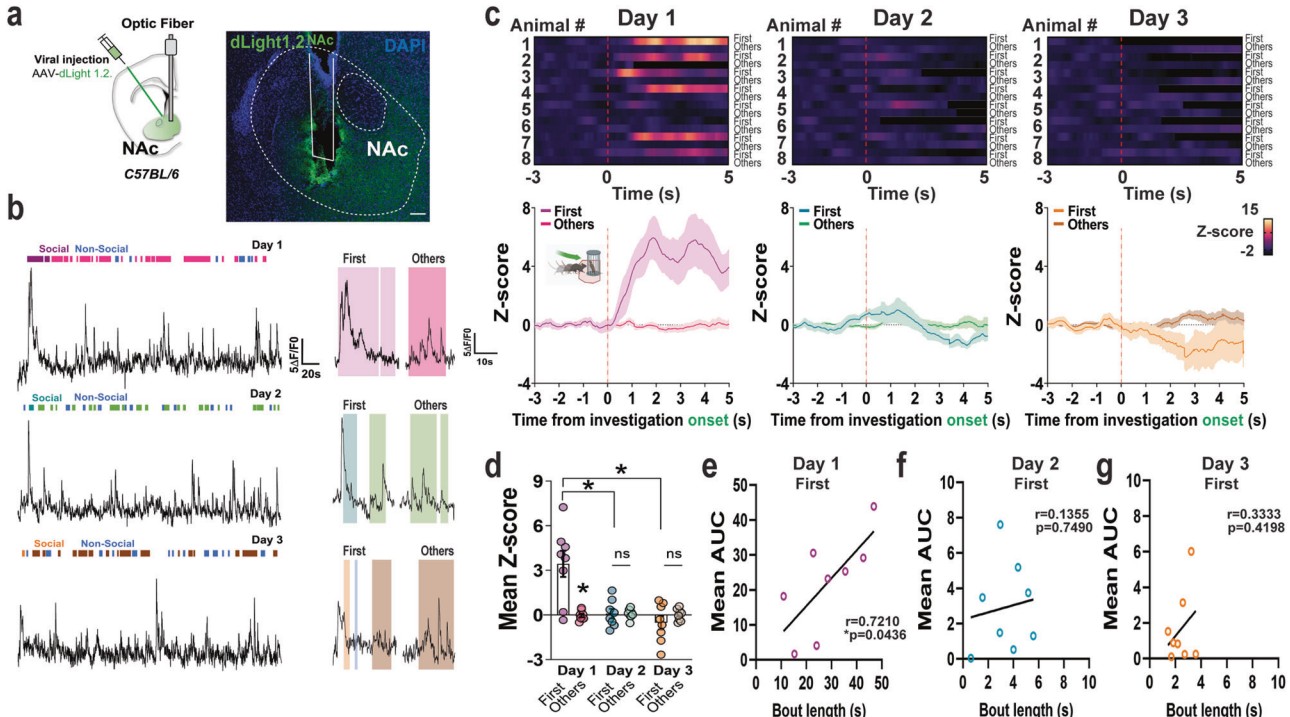

**Fig. 2 | NAc DA release encodes initial response to social novelty and bout length. a** Schematics and representative image of dLight viral-mediated expression (green signal) in the NAc of C57BL/6 mice. Nuclei are counterstained with DAPI (blue) (n = 8 male mice). Scale bar 100 μm. **b** Representative traces of NAc dLight activity recordings (dF/F0) during the first 3 days of the social paradigm. **c** Heatmap representations and z-score values of time-locked NAc DA signals relative to the time initiating a social investigation (red line) on Days 1 to 3 of the social test. The first social investigation is compared to the subsequent ones (n = 8 male mice). Inset schematic generated by Biorender. **d** Quantification of activity responses in

(**c**). Two-way RM ANOVA (time main effect: $F_{(2,14)}$ = 8.348, P = 0.012; bout main effect: $F_{(1,7)}$ = 12.84, P = 0.0089; interaction: $F_{(2,14)}$ = 10.94, P = 0.0047). Šidák's multiple comparisons *p < 0.05. Correlation between the bout length duration (s) of the first social investigation and the AUC of NAc DA signals (**e**) on Day 1, Two-tailed Pearson r (r = 0.721, $R^2$ = 0.520, p = 0.0436), (**f**) on Day 2, Two-tailed Pearson r (r = 0.135, $R^2$ = 0.018, p = 0.749) and, (**g**) on Day 3, Two-tailed Pearson r (r = 0.333, $R^2$ = 0.111, p = 0.4198) of the social test. Data represent mean ± SEM. Source data are provided as a Source Data file.

(Fig. S1c, S2e) with a longer peak latency (Fig. S1d, S2f) compared to the first novel interaction. Interestingly, the magnitude of DAergic VTA responses showed a positive correlation with the interaction bout length on the first day of social investigation (Fig. 1f, S2g). When mice investigated social familiar stimuli across days 2 and 3, overall VTA DAergic neuron responses further decreased (Fig. 1c–e, S1b and S2b–d) and correlation with bout duration reduced (Fig. 1g, h, S2h, i). Investigations of the non-social cylinder also increased VTA DAergic neuron activity although with smaller amplitude (Fig. S1e, f) and presented a delayed peak onset of activation as compared to social investigations (Fig. S1g) with shorter peak latencies (Fig. S1h). Asocial behaviors, such as self-grooming, increased VTA Daergic neuronal activity but the magnitude of the response did not habituate (Fig. S1I, j) or shift to the left (Fig. S1k) across days, demonstrating stability of the photometry recordings and excluding signal photobleaching.

To determine whether VTA DAergic neuronal responses parallel the release of DA in the nucleus accumbens (NAc), a hub for the integration of social novelty signals[9,18], we recorded NAc DA dynamics in freely behaving mice using the genetically encoded DA sensor dLight1.2[19]. Briefly, we virally expressed, in the NAc of C57BL/6 J mice, the dLight1.2 biosensor which enables ultrafast optical DA recordings and implanted an optic fiber targeting the injection site three weeks post-viral transduction (Fig. 2a, S3a, S4a). Fluctuations in NAc DA signals were recorded during the 3-chamber sociability task as in Fig. 1a. On day 1 of the test, the first investigation of novel social cues significantly increased the release of DA in the NAc although this rapidly habituated within the same session in male (Fig. 2b–d, S3b) and female mice (Fig. S4). As novel social cues became familiar across days, NAc DA signals decreased (Fig. 2b–d, S3b) and had earlier onsets of

activation (Fig. S3c) with shorter latencies (Fig. S3d). Similar to VTA Daergic neuronal dynamics, the magnitude of NAc DA release showed a positive correlation with social interaction bout length for the first social investigation on Day 1 (Fig. 2e) but not on Day 2 (Fig. 2f) or Day 3 (Fig. 2g). Overall, NAc DA release was higher during investigations of the social cylinder as compared to the non-social one (Fig. S3e, f), particularly on Day 1, and had earlier onsets of activation (Fig. S3g) with longer peak latencies (Fig. S3h). Self-grooming events also increased NAc DA release although these responses did not habituate across days (Fig. S3i, j) and did not exhibit a shift in responding (Fig. S3k).

### VTA DA activity and NAc DA neurotransmission underlies NP via modulation of length of novel social interaction

To determine if activity of VTA DAergic neurons is similarly engaged by novel social stimuli during NP, we measured VTA DAergic activity using GCaMP expression and fiber photometry as above when the mice had a choice of interaction between a novel or familiar conspecific. During the NP test (Fig. 3a), mice showed a "ping-pong" exploratory behavior between the novel and familiar social cylinders. However, the total investigation time with the cylinder containing novel social information was higher as compared to the familiar (Fig. 3b). Importantly, mice investigated the novel and familiar social cylinders with similar frequency (Fig. 3c), but the interaction bouts with familiar stimuli were shorter compared to novel stimuli (Fig. 3d), indicating that increased novelty exploration and a positive NP ratio, results from mice engaging longer investigation bouts rather than approaching more times to novel conspecifics. Here, we observed that VTA DAergic neuronal activity was significantly higher in response to novel social stimuli as compared to familiar both in males (Fig. 3e, f) and female mice (Fig. S5a, b).

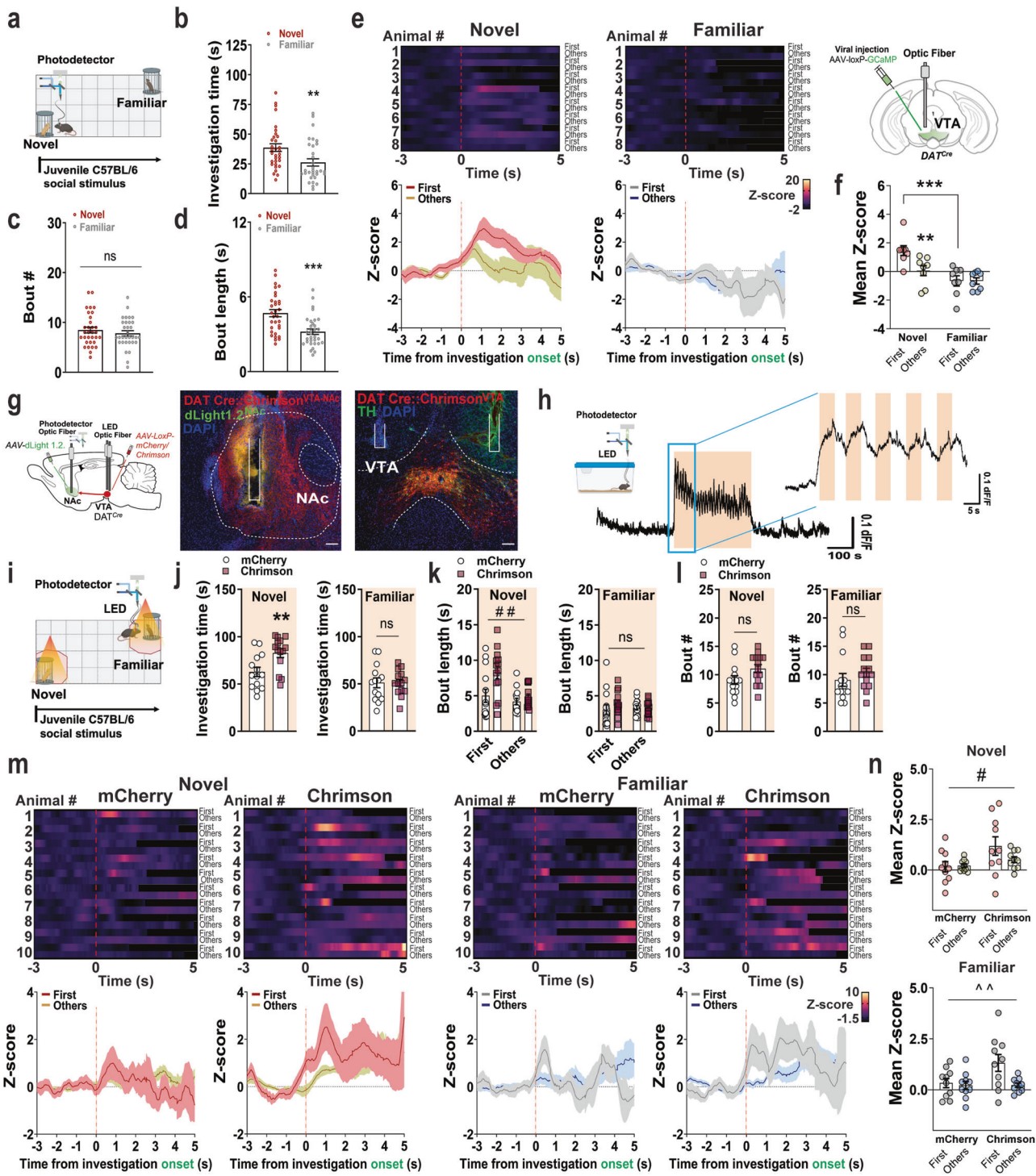

Overall, VTA DAergic activity during NP was higher and longer lasting for novel social stimuli, reinforcing the idea that increased VTA DAergic neuron activity reflects prolonged or persistent social bouts.

To test this hypothesis, we sought to optically stimulate VTA DAergic neurons during NP while simultaneously measuring NAc DA and behavior. For this experiment, DAT^Cre mice received an injection of AAV5 Cre-dependent Chrimson or control mCherry in the VTA while AAV5-dLight1.2 was delivered into the NAc region (Fig. 3g, S5c). Three weeks post-viral injection, optic fibers were implanted both in the VTA and NAc for photostimulation and photometry recordings, respectively. In mice in their home cages, light-photostimulation (593 nm, 30 Hz, 8 ms, 5 s ON- 5 s OFF) elicited reliable increases in NAc DA release (Fig. 3h). During the NP test, photostimulation of VTA DA cell bodies occurred in a closed-loop mode with light ON when mice approached the cylinders housing novel or familiar social stimuli (Fig. 3i and "Methods"). Photostimulation of VTA DAergic neurons increased the investigation time for novel but not familiar social stimuli (Fig. 3j). This was a result of prolonged bout length during novel investigations (Fig. 3k) rather than increased bout number (Fig. 3l). Escalations in novelty bout length explorations were concomitant to potentiated DA release in the NAc only for novel but not familiar social investigations (Fig. 3m, n). Altogether, these results indicate that mesolimbic DA responses contribute to NP via increasing bout length of novel social interactions.

**Fig. 3 | Photostimulation of VTA DAergic neurons promotes social novelty by increasing NAc DA release. a** NP experimental paradigm. **b** Investigation time, unpaired two-tailed $t$ test ($t_{(62)} = 2.820$, $P = 0.0064$), (**c**) bout number ($t_{(62)} = 0.8322$, $P = 0.4085$), and (**d**) bout length ($t_{(62)} = 4.133$, $P = 0.0001$) of novel and familiar conspecifics during NP ($n = 32$ male mice). **e** Left, heatmap representations and $z$-scores of VTA GCaMP signals time-locked to novel (*left*) and familiar (*right*) investigations. *Right*, VTA viral-injection and recording strategy ($n = 8$ male mice). **f** Activity responses quantifications in (**e**). Two-way RM ANOVA (novel vs familiar effect: $F_{(1,14)} = 12.96$, $P = 0.0029$; bout effect: $F_{(1,14)} = 9.279$, $P = 0.0087$; interaction: $F_{(1,14)} = 9.011$, $P = 0.0095$). **g** *Left*, schematics of virus and fiber used. *Middle*, NAc dLight expression (green) and fiber placement in DAT$^{Cre}$ mice expressing Chrimson-mCherry from VTA DAergic terminals (red). *Right*, Chrimson expression (red) with TH staining (green) and bilateral fiber placement in the VTA of the same animal. Nuclei counterstained with DAPI. Scale bars 100 μm. **h** Schematics and NA DA trace recording of a mouse receiving VTA Chrimson photostimulation. **i** Schematic of social behavior NP task closed-loop optogenetics experiment ($n = 14$ male mice/group). **j** Time (s) of social investigations. *Left*, novel investigations two-tailed $t$ test ($t_{(26)} = 2.950$, $P = 0.0066$), *right*, familiar investigations two-tailed $t$ test ($t_{(26)} = 0.01689$, $P = 0.9867$). **k** Bout length (s) of novel

(*left*) investigations two-way RM ANOVA (virus effect: $F_{(1,26)} = 8.589$, $P = 0.0070$; bout effect: $F_{(1,26)} = 8.234$, $P = 0.0081$; interaction: $F_{(1,26)} = 3.198$, $P = 0.0854$) and familiar (*right*) investigations two-way RM ANOVA (virus effect: $F_{(1,26)} = 0.5525$, $P = 0.4640$; bout effect: $F_{(1,26)} = 0.1767$, $P = 0.6777$; interaction: $F_{(1,26)} = 1.016$, $P = 0.3227$). **l** Novel bout number (*left*) unpaired two-tailed t test ($t_{(26)} = 1.948$, $P = 0.0628$) and familiar (*right*) investigations unpaired two-tailed t test ($t_{(26)} = 0.9405$, $P = 0.3556$). **m** Heatmap representations and $z$-scores of NAc DA signals time-locked to novel (*left panels*) and familiar (*right panels*) conspecific investigations in DAT$^{VTA:mCherry}$ and DAT$^{VTA:Chrimson}$ mice ($n = 10$ male mice/group). **n** Quantification of NAc DA responses shown in (**m**). *Top*, novel investigations, two-way RM ANOVA (virus effect: $F_{(1,18)} = 5.154$, $P = 0.0357$; bout effect: $F_{(1,18)} = 1.385$, $P = 0.2545$; interaction: $F_{(1,18)} = 1.916$, $P = 0.1832$). *Bottom*, familiar investigations, two-way RM ANOVA (virus effect: $F_{(1,18)} = 3.583$, $P = 0.0746$; bout effect: $F_{(1,18)} = 5.442$, $P = 0.0315$; interaction: $F_{(1,18)} = 3.535$, $P = 0.0764$). Schematics in (**a**), (**h**), and (**i**) generated in Biorender. Data represent mean ± SEM. Two-way RM ANOVA #p < 0.05, ##p < 0.01, ^^p < 0.01. Unpaired two-tailed $t$ test/Šidák's multiple comparisons **p < 0.01, *** p < 0.001. Source data are provided as a Source Data file.

## IPN GAD2 neurons are inhibited by social novelty and activated during termination of familiar social investigations

Previously, we demonstrated that IPN GABAergic neurons represent a neuroanatomical substrate for familiarity signaling and NP[15], although the real-time neural circuit dynamics causally involved in these responses are unknown. Here, we recorded IPN GABAergic neuronal activity during NP as in Fig. 3a. Mice bearing one copy of Cre recombinase under the glutamic acid decarboxylase 2 (GAD2) enzyme promoter were injected with Cre-dependent GCaMP in the IPN via AAV5-mediated gene delivery (Fig. 4a, S6a). Three weeks after viral injection, an optic fiber was implanted in the ventral caudal IPN which allows for neuronal recording within the IPN, directly, reducing potential signals from GABAergic neurons within the neighboring VTA. During the NP paradigm, activity of IPN GAD2 neurons were inhibited during investigation of novel social stimuli and were higher in response to familiar social stimuli as compared to novel (Fig. 4b, c), opposite to what we observed for mesolimbic DA activity. When mice investigated novel conspecifics, IPN GAD2 neuronal inhibition was long-lasting as compared to familiar explorations. Interestingly, some DA responses to novelty are not engaged with the onset but instead, with the retreat of exploratory events, which is necessary to reinforce the avoidance of potential threats[20]. Considering that the IPN has been linked to avoidance of aversive stimuli[16], we time-locked activity dynamics with the offset of social investigations. Importantly, we found significantly larger increases of IPN GABAergic neuronal activity with the termination of familiar social investigations, as compared to the novel (Fig. 4d–f). As mentioned above, in the NP task, a positive NP ratio results from mice engaging longer investigation bouts to novel conspecifics. Based on this fundamental observation, social NP should be tightly regulated by circuits that control the bout duration to conspecific interactions. If IPN GABAergic neuronal activity is engaged with the termination of familiar conspecific investigations to constrict social novelty responses, these neurons may be key in determining bout length durations and therefore a positive NP ratio. We previously demonstrated that photoinhibition of IPN GAD2 neuronal cell bodies increased familiar social investigations during the NP task[15]. Here, we performed a deeper reanalysis of our previous results and found that silencing IPN GAD2 activity during social NP[15] (Fig. S6b) resulted in prolonged familiar interaction bouts (Fig. S6c) rather than increased number of exploratory events (Fig. S6d).

## IPN→LDTg GAD2 projecting neurons are inhibited by social novelty and encode the offset of familiar social investigations

The IPN is a midbrain structure that sends axonal projections to forebrain and hindbrain areas[21,22]. To explore whether IPN GAD2 neurons are local interneurons or inhibitory projecting neurons, we used

anterograde viral-mediated circuit tracing. GAD2$^{Cre}$ mice were injected with AAV2-Cre-dependent eGFP into the IPN (Fig. 4g) and a wide range brain histological analysis was performed three weeks post-viral injection. Axon terminals from IPN GAD2 neurons were detected in the median and dorsal raphe (MnR, DR)(Fig. S6e, f), and hindbrain structures including the LDTg (Fig. 4h), the interfascicular part of the DR (DRI), the nucleus incertus and the central pontine gray (CGPn) (Fig. S6g), as previously described for GABAergic IPN neuronal populations[23]. In LDTg slices, IPN axon terminals colocalized with the presynaptic markers synaptophysin (Fig. S6h) and the vesicular inhibitory amino acid transporter (VIAAT) (Fig. 4i), suggesting these are inhibitory terminals rather than fibers of passage. To verify the IPN$^{GAD2}$→LDTg circuit, we co-injected AAV2-hSyn-mCherry virus with retrograde tracing AAVrg-FLEX-eGFP bilaterally in the LDTg (Fig. 4j) and imaged the IPN. GAD2 neurons from the IPN→LDTg circuit localized to the ventral and caudal IPN, but not the rostral part (Fig. 4k).

To record the activity of LDTg-projecting IPN GABAergic cell bodies, we injected retrograde virus containing Cre-dependent GCaMP bilaterally into the LDTg of GAD2$^{Cre}$ mice and four weeks post-viral injection we implanted an optic fiber at the ventral-caudal IPN (Fig. 5a, S6i). Similar to overall activity of IPN GAD2 neurons, during NP the IPN$^{GAD2}$→LDTg circuit was robustly inhibited when mice investigated novel social stimuli, but not familiar (Fig. 5b, c). Remarkably, increases in IPN$^{GAD2}$→LDTg circuit activity were stronger with the termination of familiar social investigations as compared to novel (Fig. 5d–f). Altogether, these results indicate that inhibition of the IPN$^{GAD2}$→LDTg neuronal activity may be necessary for prolonged investigation of social novelty behavior; whereas, increases in activity may underlie termination of social investigations when these are no longer novel.

## Termination of familiar social interactions during social NP requires IPN→LDTg circuit activity

To test the requirement of the IPN$^{GAD2}$→LDTg circuit during a social choice paradigm, GAD2$^{Cre}$ mice received a bilateral injection of AAV2-retro Cre-dependent halorhodopsin (NpHR3.0) in the LDTg (Fig. S7a) and four weeks post injection an optic fiber cannula was implanted targeting the ventral-caudal IPN region (Fig. 5g, S7b). Control mice were injected into the LDTg with AAV-retro Cre-dependent eGFP. During the NP task, silencing of the IPN$^{GAD2}$→LDTg circuit occurred in a closed-loop mode coupled to social explorations (Light-On when mice entered the area surrounding the cylinders containing novel and familiar stimuli, Fig. 5h and Methods). Optogenetic photoinhibition (593 nm, constant) of the IPN$^{GAD2}$→LDTg circuit significantly increased the investigation time to the familiar social stimulus with no effect on novel social stimuli investigations (Fig. 5i), thereby disrupting the NP ratio (Fig. 5j). Akin to overall IPN GAD2 silencing, IPN$^{GABA}$→LDTg

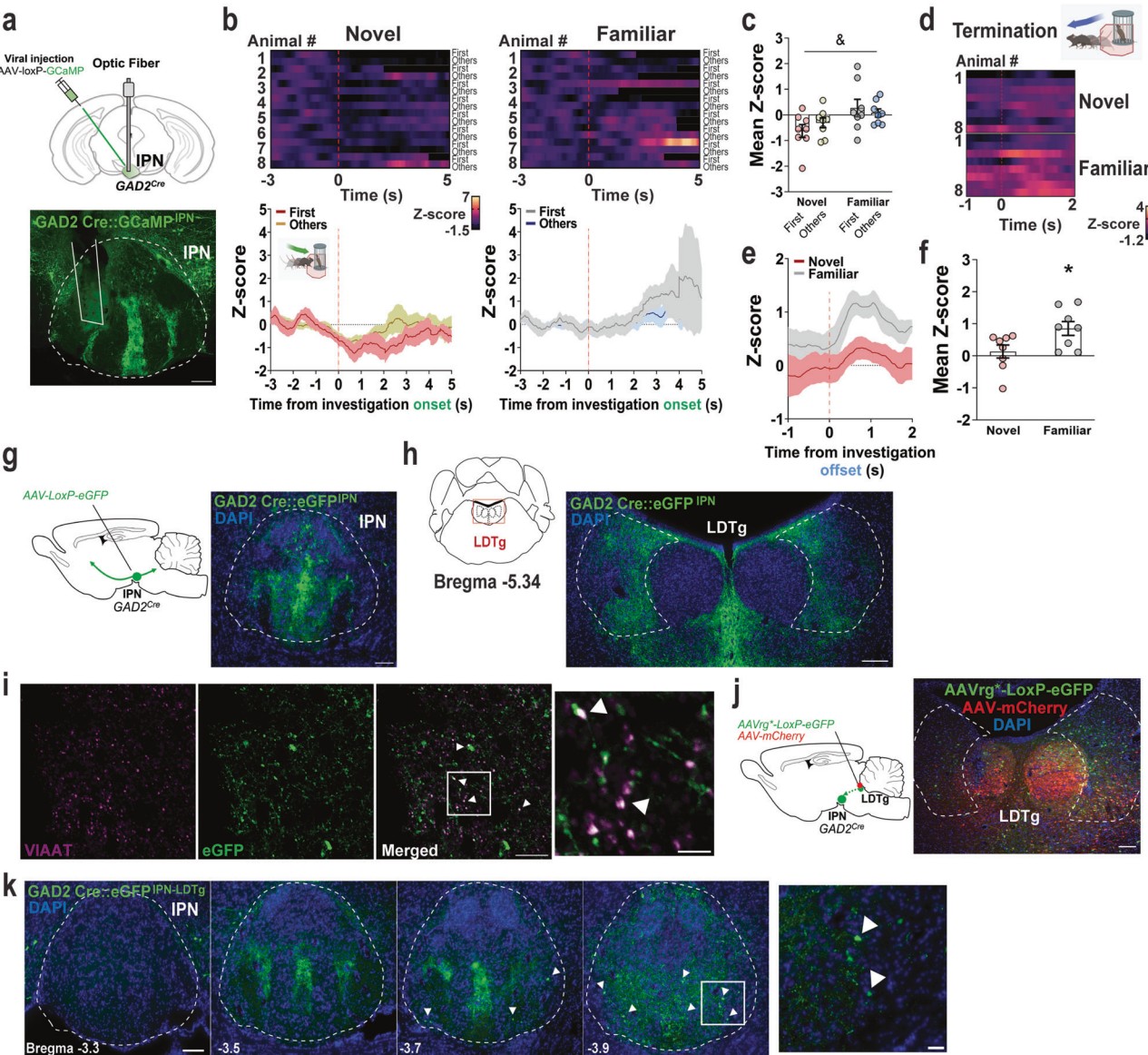

**Fig. 4 | IPN GAD2 activity is inhibited by social novelty and ramps up with the termination of familiar social investigations. a** *Upper panel*, schematics and representative image of viral-mediated GCaMP expression in the IPN of GAD2 Cre mice. Scale bar 100 μm. *Lower panel*, schematic of fiber placements from the recorded animals. **b** Heatmap representations and *z*-score values of IPN GAD2 neuronal activity time-locked to investigations of novel (*left panels*) and familiar (*right panels*) conspecifics on Day 4 of the social NP test (n = 8 male mice). **c** Quantification of activity responses in (**b**). Two-way RM ANOVA (novel vs familiar main effect: $F_{(1,14)} = 4.926$, $P = 0.0435$; bout main effect: $F_{(1,14)} = 0.1097$, $P = 0.7454$; interaction: $F_{(1,14)} = 1.795$, $P = 0.2017$). **d** Heatmap representations (inset schematic generated with Biorender) and (**e**) *z*-score values of time-locked IPN GAD2 neuronal activity relative to the offset (s) of social investigation (red line) during the NP test (n = 8 male mice). **f** Quantification of activity responses in (**e**). Unpaired two-tailed *t* test ($t_{(14)} = 2.415$, $P = 0.03$). **g** Schematics and representative image of viral-mediated eGFP expression in the IPN of GAD2$^{Cre}$ mice. Nuclei are counterstained with DAPI. Scale bar 100 μm. **h** Axonal projections from IPN GAD2 neurons innervating the laterodorsal tegmental (LDTg) area. Nuclei are counterstained with DAPI. Scale bar 100 μm. **i** Immunostaining of the vesicular inhibitory amino acid transporter (VIAAT, magenta) in the LDTg, colocalized (arrows) with eGFP + axon terminals from IPN GAD2 neurons (green). Scale bar 20 μm and 5 μm (far right panel). **j** Schematic of IPN→LDTg neuron retro-labeling and photomicrograph of the LDTg injection site. Nuclei are counterstained with DAPI. Scale bar 100 μm. **k** Representative images of retrograde labeled IPN GAD2 neurons from the LDTg (arrows). Nuclei are counterstained with DAPI. Scale bar 100 μm and 25 μm (far right panel). Data represent mean ± SEM. Two-way and one-way RM ANOVA & p < 0.05. Unpaired two-tailed *t* test *p < 0.05. Source data are provided as a Source Data file.

photoinhibition increased social bout length during familiar but not novel social stimuli investigations (Fig. 5k), without affecting bout number (Fig. 5l), suggesting that mice were unable to terminate familiar explorations when the IPN$^{GABA}$→LDTg circuit is photoinhibited. Total exploratory behavior was similar between the two groups excluding gross changes in locomotor activity (Fig. 5m). Injection of AAV2-retro Cre-dependent NpHR3.0 in the DR, a region close to the LDTg, did not affect the investigation time for novel or familiar interactions or the social NP ratio (Fig. S7c–g). Together, these results

indicate that increased IPN$^{GAD2}$→LDTg circuit activity observed with the termination of familiar social investigations acts as a brake on lengthy novel exploratory events to control social NP responses.

## LDTg ChAT + neurons projecting to the VTA receive GABAergic inputs from the IPN

The LDTg sends strong cholinergic excitatory inputs to the VTA DA system, and these projections have been implicated in DAergic neuron firing rates and reward-related behavior[24,25]. While attempts have been

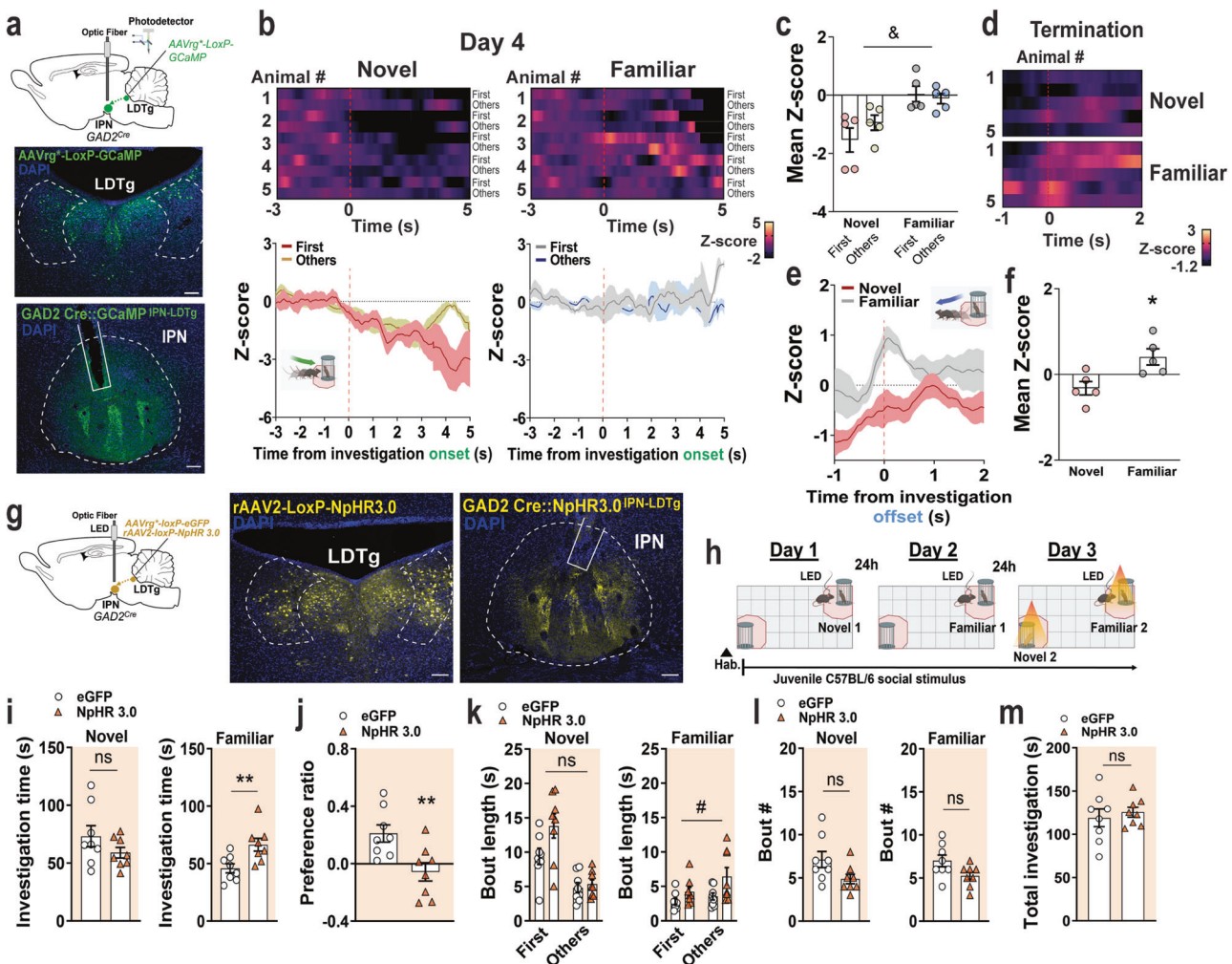

**Fig. 5 | IPN outputs to the LDTg convey inhibitory signals by social novelty and the termination of familiar social investigations. a** *Top*, injection and optic fiber implant strategy. *Middle*, representative image of retroviral-mediated GCaMP injection in the LDTg of GAD2$^{Cre}$ mice. *Bottom*, photomicrograph of retroviral GCaMP-expressing IPN→LDTg neurons and fiber track. Nuclei are counterstained with DAPI. Scale bars 100 µm. **b** Heatmap representations and *z*-scores of IPN→LDTg GAD2 neuronal activity time-locked to novel (*left*) and familiar (*right*) investigations on social NP Day 4 (n = 5 male mice). **c** Quantification of activity responses in (**b**). Two-way RM ANOVA (novel vs familiar main effect: $F_{(1,8)}$ = 10.11, $P$ = 0.0130; bout main effect: $F_{(1,8)}$ = 1.858, $P$ = 0.2099; interaction: $F_{(1,8)}$ = 5.486, $P$ = 0.0473). **d** Heatmap representations (n = 5 male mice/group) and (**e**) z-scores of time-locked IPN→LDTg GAD2 neuronal activity relative to the offset (s) of social investigation (red line). **f** Quantification of activity responses in (**e**). Unpaired two-tailed *t* test ($t_{(8)}$ = 2.992, $P$ = 0.0173). **g** Schematics of the retroviral injection and optic fiber implant strategy (*left*). Representative image of retroviral-mediated NpHR3.0:eYFP injection in the LDTg of GAD2$^{Cre}$ mice (*middle*). Photomicrograph of retroviral NpHR3.0:eYFP-expressing IPN→LDTg neurons and IPN fiber track (*right*). Nuclei are counterstained with DAPI. Scale bars 100 µm. **h** Schematic of the social

NP task with closed-loop optogenetic photoinhibition during the NP test. **i** Time (s) of novel and familiar social investigations in GAD2$^{IPN→LDTg:eGFP}$ and GAD2$^{IPN→LDTg:NpHR}$ mice during the NP task. (n = 8 male mice/group). *Left*, novel investigations unpaired two-tailed t test ($t_{(14)}$ = 1.394, $P$ = 0.1851). *Right*, familiar investigations unpaired two-tailed *t* test ($t_{(14)}$ = 3.055, $P$ = 0.0086). **j** Social NP. Unpaired two-tailed *t* test ($t_{(14)}$ = 3063, $P$ = 0.0084). **k** Interaction bout length (s) of social investigations. *Left*, novel investigations two-way RM ANOVA (virus main effect: $F_{(1,14)}$ = 3.358, $P$ = 0.0882; bout main effect: $F_{(1,14)}$ = 47.59, $P$ < 0.0001; interaction: $F_{(1,14)}$ = 4.308, $P$ = 0.057). *Right*, familiar investigations two-way RM ANOVA (virus main effect: $F_{(1,14)}$ = 5.159, $P$ = 0.0394; bout main effect: $F_{(1,14)}$ = 5.24, $P$ = 0.0381; interaction: $F_{(1,14)}$ = 1.387, $P$ = 0.2586). **l** Social bout number. *Left*, novel investigations unpaired two-tailed t test ($t_{(14)}$ = 2.076, $P$ = 0.0567). *Right*, familiar investigations unpaired two-tailed t test ($t_{(14)}$ = 2.139, $P$ = 0.0506). **m** Total investigation time (s) during the NP task. Unpaired two-tailed t test ($t_{(14)}$ = 0.5520, $P$ = 0.5897). **b**, **e** insets and (**h**) schematic generated with Biorender. Data represent mean ± SEM. Two-way RM ANOVA &p < 0.05, #p < 0.05. Unpaired two-tailed t test/Šidák's multiple comparisons *p < 0.05, ** p < 0.01. Source data are provided as a Source Data file.

made to correlate IPN function with reward-related information[26,27], there are scant data indicating that IPN activity directly affects meso-limbic DA neurotransmission. We observed that axon terminals from IPN GAD2$^{Cre}$ injected mice enclosed LDTg cholinergic neurons (Fig. S8a). Thus, we investigated whether the IPN uses LDTg cholinergic neurons as a bridge to modulate VTA function. First, we verified the IPN is synaptically connected to LDTg cholinergic neurons that innervate the VTA. To this aim, we injected the IPN of C57BL/6 mice with AAV1-Cre virus which moves transynaptically to postsynaptic cells, and injected Cre-dependent eGFP in the VTA of the same animal

(Fig. 6a). We found eGFP + cell bodies in the LDTg that colocalized with choline acetyltransferase (ChAT, Fig. 6b, c), suggesting the IPN inner-vates cholinergic LDTg neurons that project to the VTA. Next, to verify the IPN neurons establish direct synaptic connectivity with LDTg neurons innervating the VTA, we performed electrophysiological recordings in acute mouse brain slices. We injected a Cre-dependent mCherry tag packaged into a retrograde viral vector (AAVrg-DIO-mCherry) into the VTA of 4-week-old mice expressing Cre recombinase under the ChAT promoter (ChAT$^{Cre}$) while injecting channelrhodopsin (ChR2, AAV5-ChR2-eYFP) into the IPN (Fig. 6d). Three weeks later, we

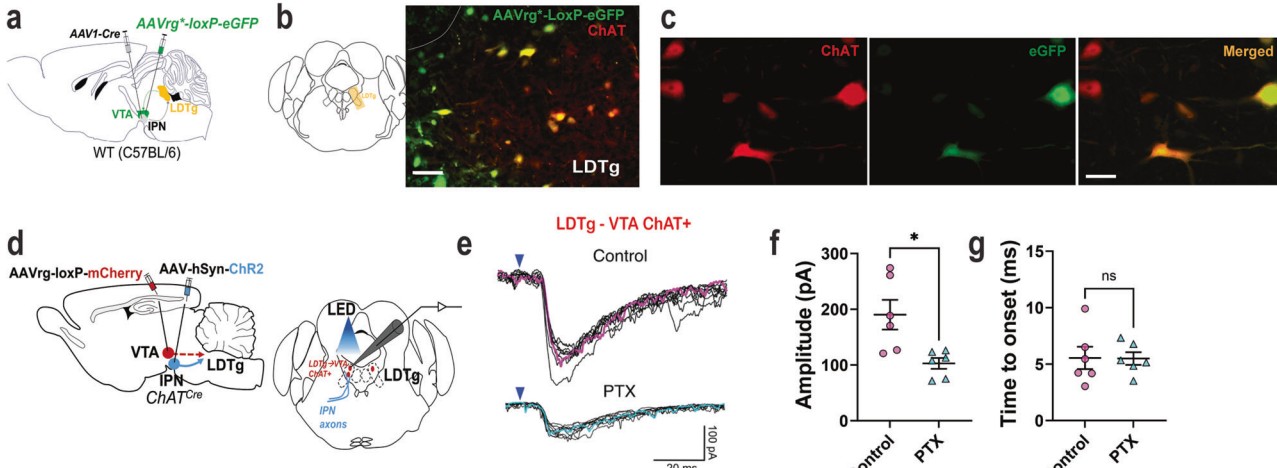

**Fig. 6 | The IPN inhibits LDTg cholinergic neurons that innervate the VTA.**
**a** Schematics of AAV1-Cre injection in the VTA and retro-viral loxP-eGFP injection in the IPN of C57BL/6 mice. **b** Representative image of viral-mediated eGFP expression (green) in the LDTg immunostained with the ChAT marker (red). Scale bar 50 μm. **c** Magnified view of LDTg ChAT + neurons (*left*), viral-mediated eGFP + expression (*middle*) and their colocalization (*right*). Scale bar 20 μm. **d** Schematics of the viral strategy and recording strategy used to measure LDTg→VTA ChAT + responses upon IPN axon terminal photostimulation. **e** Representative current traces of LDTg→VTA ChAT + neuronal responses upon blue light photostimulation (450 nm,

blue arrow) of IPN axon terminals recorded before (Control) and during exposure to picrotoxin (PTX). The average of 10 traces is represented by magenta (control) and turquoise (PTX) lines within each group. **f** Average of postsynaptic current peak amplitude (pA) in control and PTX conditions (n = 3 animals, 6 cells), unpaired two-tailed t test ($t_{(5)}$ = 3.410, $P$ = 0.019). **g** Average of IPSCs response onset (ms) in the control and PTX groups after blue light (450 nm) stimulation, unpaired two-tailed $t$ test ($t_{(5)}$ = 0.0881, $P$ = 0.933). Data represent mean ± SEM. *p < 0.05. Source data are provided as a Source Data file.

patched LDTg-mCherry + neurons and stimulated IPN afferents with blue light (i.e., 470 nm) to evoke synaptic currents. In control conditions, with 25 μM NBQX and 30 μM d-AP5 in the bath to block AMPA and NMDA receptors respectively (Fig. 6e, top panel), light stimulation evoked inward currents at −90 mV with amplitudes decreasing as the holding potential became less negative before reversing at −45 + /− 3 mV (Fig. S8b, c), matching the theoretical reversal potential for Cl⁻ in our experimental conditions (45.13 mV with [Cl-]$_i$ = 20 mM and room temperature of 20°C). To confirm that these synaptic events were GABAergic we applied 50 μM picrotoxin in the bath. In the presence of the blocker, IPSC amplitudes significantly decreased by ~46% compared to control conditions (Fig. 6e, f, S8b, c). Of note, the picrotoxin-resistant current also reversed at the reversal potential for Cl⁻ (Fig. S8b, c). We also found that the onset of synaptic currents was about 5 ms (Fig. 6g), a value indicative of monosynaptic responses. Together, these results suggested the IPN may use LDTg ChAT + neurons to regulate mesolimbic DA transmission.

## Activation of the IPN→LDTg circuit during social NP reduces length of social novelty exploration by decreasing NAc DA release

To test the functional role of the IPN^GAD2→LDTg circuit on social novelty responses and the release of NAc DA, we selectively expressed Cre-dependent Chrimson in IPN GABAergic neurons and photostimulated LDTg projections while we recorded real-time NAc DA release. To this aim, GAD2^Cre mice received dual injections of viral particles containing either mCherry or Chrimson plasmids into the IPN and the dLight1.2 biosensor into the NAc region (Fig. 7a). Three weeks post-viral infusion, a bilateral optic fiber cannula was implanted in the LDTg to photostimulate IPN axons innervating this region and a second optic fiber implant targeted the NAc for photometry recordings (Fig. 7a and S9a). First, we tested IPN^GAD2→LDTg circuit photostimulation (593 nm, 20 Hz, 12 ms/pulse) in mice within their home cages and found that light delivery significantly reduced the number of NAc DA peaks (Fig. S8d, e), suggesting this circuit can influence NAc DA release. Next, in the social novelty test, light photostimulation (593 nm, 20 Hz, 12 ms/ pulse) was delivered in closed-loop mode when animals entered the

area nearby the cylinders during the social NP task, as in Fig. 3i. Photostimulation of the IPN^GAD2→LDTg circuit reduced novel social investigation while no effect was observed in familiar social investigations (Fig. 7b). As a result, the NP ratio was significantly impaired (Fig. 7c). Photostimulation of the IPN^GAD2→LDTg circuit reduced the bout length duration for novel but not familiar explorations (Fig. 7d), without affecting the total bout number of exploratory events (Fig. 7e). Total social investigation was similar between groups (Fig. 7f), excluding overall locomotor activity disturbances. In addition, during novel social investigations the release of DA in the NAc was significantly reduced in GAD2^IPN→LDTg:Chrimson mice compared to GAD2^IPN→LDTg:mCherry mice (Fig. 7g, h). No significant effect was detected during familiar social explorations (Fig. 7g, h). Furthermore, photostimulation of IPN GAD2 neuronal cell bodies that project to the LDTg area during the social novelty choice paradigm (Fig. S9b−g) also reduced social novelty exploration but not familiar investigations (Fig. S9d). Indeed, photoactivation of IPN GAD2 projecting neurons disrupted the NP ratio (Fig. S9e) without affecting total amounts of exploratory behavior (Fig. S9f). Together these data indicate that increasing activation of the IPN→LDTg circuit during social NP specifically reduces expression of social novelty behaviors by constraining the bout length of social novelty exploratory events and their associated NAc DA release.

## Discussion
Individuals adopt strategic patterns of exploratory behavior when presented to new stimuli that fine-tune with multiple contacts once the stimulus becomes familiar[17,20]. These optimal investigative events depend on the salience of the new stimuli[28], are preferred over familiar stimuli, and are critical to guide appropriate social responses and NP that maintain rates of sociability across multiple species[4]. In the present study we combined fiber photometry recordings, optogenetic manipulations, and behavioral assays to unravel behavioral sequences and circuitry dynamics guiding social novelty responses and NP.

Our results affirm that social interactions bear rewarding aspects and recruit neural circuits of motivation[8,29], including VTA DAergic pathways[9−12,30−32]. We observed larger VTA signals occurring with the very first bout of exploration which rapidly habituated with

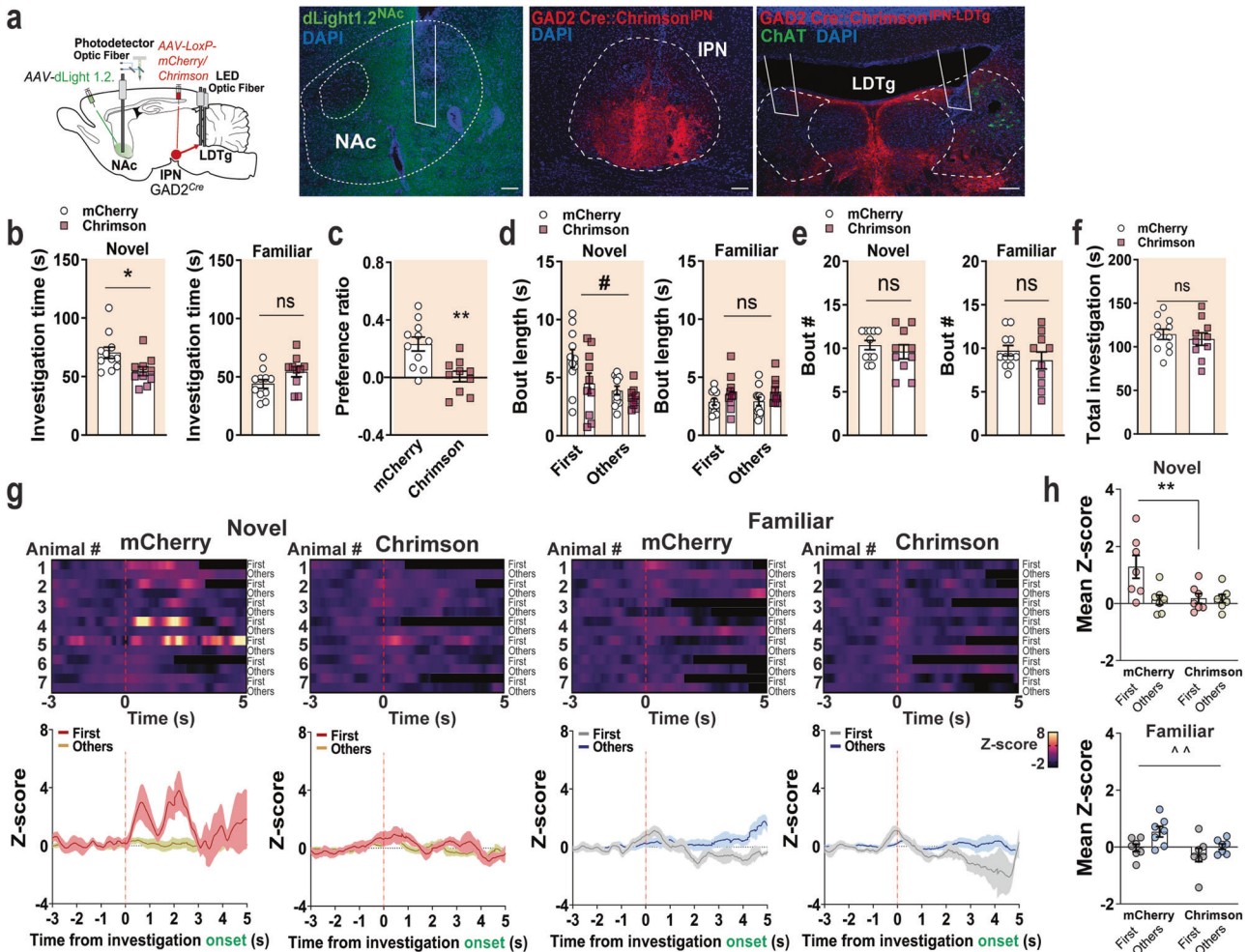

**Fig. 7 | The IPN^GAD2 → LDTg circuit controls social NP by modulating the levels of NAc DA. a** Schematics of the viral injection and optic fiber implant strategy used (*left*). Photomicrograph of dLight1.2 expression and fiber track in the NAc. Representative images of Chrimson injection in the IPN of GAD2^Cre mice and bilateral fiber implant in the LDTg to photostimulate IPN GAD2 + axons. ChAT immunostaining (green) identifies cholinergic neurons in the LDTg area. Nuclei are counterstained with DAPI. Scale bars, 100 μm. **b** Time (s) of novel and familiar social investigations in GAD2^IPN→LDTg:mCherry (n = 11 male) and GAD2^IPN→LDTg:Chrimson (n = 10 male) mice during the NP task. *Left*, novel investigations unpaired two-tailed t test ($t_{(19)} = 2.537$, $P = 0.0201$). *Right*, familiar investigations unpaired two-tailed t test ($t_{(19)} = 1.857$, $P = 0.0789$). **c** Social NP ratio in GAD2^IPN→LDTg:mCherry and GAD2^IPN→LDTg:Chrimson mice. Unpaired two-tailed t test ($t_{(19)} = 3.633$, $P = 0.0018$). **d** Bout length (s) of social investigations in GAD2^IPN→LDTg:mCherry and GAD2^IPN→LDTg:Chrimson mice during the NP test. *Left*, novel investigations two-way RM ANOVA (virus main effect: $F_{(1,19)} = 5.118$, $P = 0.0356$; bout main effect: $F_{(1,19)} = 9.326$, $P = 0.0065$; interaction: $F_{(1,19)} = 1.314$, $P = 0.2660$). *Right*, familiar investigations two-way RM ANOVA (virus main effect: $F_{(1,19)} = 3.999$, $P = 0.0600$; bout main effect: $F_{(1,19)} = 0.06706$, $P = 0.7984$;

interaction: $F_{(1,19)} = 0.0143$, $P = 0.9061$). **e** Bout number of social investigations in GAD2^IPN→LDTg:mCherry and GAD2^IPN→LDTg:Chrimson mice. *Left*, novel investigations unpaired two-tailed t test ($t_{(19)} = 0.8078$, $P = 0.4292$). *Right*, familiar investigations unpaired two-tailed t test ($t_{(19)} = 1.008$, $P = 0.3263$). **f** Total interaction time (s) in GAD2^IPN→LDTg:mCherry and GAD2^IPN→LDTg:Chrimson mice during the NP task. Unpaired two-tailed t test ($t_{(19)} = 0.5842$, $P = 0.5660$). **g** Heatmap representations and *z*-score values of NAc DA signals time-locked to investigations of novel (*left panels*) and familiar (*right panels*) conspecifics in GAD2^IPN→LDTg:mCherry and GAD2^IPN→LDTg:Chrimson mice during the social NP test (n = 7 male mice/group). **h** Quantification of the NAc DA responses in (g) when mice investigated novel (*top*), two-way RM ANOVA (virus main effect: $F_{(1,12)} = 3.993$, $P = 0.0689$; bout main effect: $F_{(1,12)} = 6.930$, $P = 0.0219$; interaction: $F_{(1,12)} = 6.798$, $P = 0.0229$) and familiar (*bottom*) social stimuli, two-way RM ANOVA (virus main effect: $F_{(1,12)} = 3.722$, $P = 0.0777$; bout main effect: $F_{(1,12)} = 10.28$, $P = 0.0076$; interaction: $F_{(1,12)} = 1.008$, $P = 0.3352$). Data represent mean ± SEM. Two-way RM ANOVA #p < 0.05, ^^p < 0.01. Unpaired two-tailed t test *p < 0.05, **p < 0.01. Source data are provided as a Source Data file.

---

subsequent events, indicating that VTA DAergic neurons are engaged by the novelty component of conspecific interactions[9,10,30]. Consistent with a recent report, similar responses were detected when we recorded NAc DA neurotransmission[33]. Interestingly, both DAergic neuron activity and accumbens DA release during the initial encounter with a novel conspecific correlated with the bout length of interaction. Importantly, during NP the biased DA signal toward the first encounter with a novel social stimuli was retained and optogenetic activation of DAergic neurons to boost accumbens DA neurotransmission resulted in increased NP by specifically increasing bout length with the novel social stimulus predominantly during the initial social encounter. These data suggest that the magnitude of accumbens DA motivates

and sustains interest towards a novel social conspecific during the very first interaction, an effect that rapidly habituates with repeated investigations. Indeed, in the NP task, the levels of VTA DA engagement and NAc DA release by social novelty hardly reached the magnitude of activation detected on day 1, suggesting a context-dependence in the salience of novel social interactions. Future experiments should test if novel investigations increase DA during an initial encounter to elicit associative learning and/or to promote future reward-oriented approach behavior relative to the salience of the stimuli[34–36].

Previously, we associated the IPN with familiarity responses and NP[15], and the current study reveals that activity of IPN neurons is dynamically regulated by the novelty of social stimuli. As opposed to

the mesolimbic DAergic system, IPN GAD2 neuronal activity decreased when animals encountered social novelty as compared to social familiarity. The IPN is a relatively small area that receives mainly excitatory input from the medial habenula[22,27]. Although the habenulo-interpeduncular axis can regulate reward-related information[37,38] and emotional drives such as anxiety or fear-related behaviors[21,39,40], it is generally assumed that overall activation of the IPN promotes avoidance states[41]. Thus, reductions of IPN activity during social novelty investigations likely prevents avoidance signals during reward-related approach behaviors. Interestingly, our data indicate that activity of IPN GAD2 neurons is recruited with the termination of familiar social investigations. Additional DA systems linked to avoidance behaviors are engaged by cessation of novel stimuli exploration presumably to reinforce the avoidance of aversive stimuli based on their novelty[20]. Assuming that familiar social stimuli bear less salience than novel stimuli, these results suggest the IPN may encode the motivation of social interactions when they are no longer salient, thus generating stop signals of learned action sequences[42].

We identified that IPN GAD2 neurons send extensive projections to the hindbrain, including the LDTg area, as previously reported for IPN neuronal subtypes[23,27]. The LDTg send abundant excitatory and inhibitory inputs to the VTA that control DA neuron burst firing and reward processing[24,43,44]. Recent work indicates the IPN inhibits LDTg neurons projecting to the VTA to control nicotine aversion[41], and our current findings demonstrate that the IPN$^{GAD2}$→LDTg circuit regulates NP by modulating DA signals. Combining viral tracing and brain slice electrophysiological recordings, we found that IPN neurons provide inhibitory monsosynaptic input to LDTg ChAT + neurons that project to the VTA. Optically evoked IPSCs in LDTg cholinergic neurons were at least partially blocked by a GABA$_A$ receptor antagonist and all inhibitory currents were carried by chloride providing a potential mechanism by which IPN inhibitory input reduces activity of cholinergic LDTg neuron that innervate the VTA to potentially reduce VTA DA activity. However, mesolimbic DA and motivational behaviors are also controlled by separate LDTg GABA and glutamatergic populations that directly or indirectly project to the VTA[44]. Thus, we cannot exclude the possibility that the IPN also innervates LDTg GABAergic and/or glutamatergic neurons to modulate aversive and motivational behaviors, a topic that requires future study.

Our fiber photometry data revealed long-lasting inhibition of the IPN$^{GAD2}$→LDTg circuit during novel social explorations and increased activity with the termination of familiar social investigations. Silencing IPN→LDTg circuit prolonged the bout length of familiar social investigations, suggesting that this circuit is necessary to reduce motivation of familiar conspecific interactions to control social NP. This effect was not observed when we photoinhibited IPN neurons that project to the DR. Importantly, we provide evidence that the IPN can directly modulate DA in the NAc in real-time, as postulated in prior studies[26]. Together, our data indicate that increasing activation of an inhibitory IPN→LDTg circuit during social NP specifically reduces expression of social novelty by constraining the bout length of social novelty exploratory events and their associated NAc DA release. These results place the IPN→LDTg pathway as an important circuit element regulating motivational behaviors associated with social NP.

In summary, our study provides unprecedented analysis on the dynamics of DA signals causally linked to social novelty where DA neurotransmission encodes length of novel social interaction during an initial encounter. Our results indicate the IPN→LDTg circuit controls NP by modulating mesolimbic DA signaling where activation of this circuit reduces DA neurotransmission thereby reducing conspecific interaction. Disrupted DA neurotransmission is linked to social impairment in numerous neurodevelopmental and psychiatric diseases[29,45]. These results open the possibility of assessing the functionality of the IPN→LDTg circuit in disorders with social motivation dysfunction, such as in autism spectrum disorders[46].

## Methods

### Animals

All experiments followed the guidelines for care and use of laboratory animals provided by the National Research Council, and with approved animal protocols from the Institutional Animal Care and Use Committee of the University of Massachusetts Chan Medical School (UMCMS). C57Bl/6J (Stock #000664, Jackson), *GAD2$^{Cre}$* (Stock #10802, Jackson), *DAT$^{Cre}$* (Stock #006660, Jackson), *ChAT$^{Cre}$* (Stock #006410, Jackson) mice were bred in the UMMS animal facility and used in behavioral, optogenetic and fiber photometry experiments as indicated. Cre lines were crossed with C57Bl/6J mice and only heterozygous animals carrying one copy of the Cre recombinase gene were used for experimental purposes. For social experiments, juvenile stimuli always consisted of sex-matched C57Bl/6J mice (4–7 weeks old) bred in the UMCMS animal facility. All mice were housed together (21.3 °C and 49.9% humidity) and kept on a standard 12 h light/dark cycle (lights ON at 7 A.M.) with *ad libitum* access to food and water. Three to four weeks before experimentation, subject mice were kept under a reverse 12 h light/dark cycle (lights ON at 7 P.M.), and individually housed for at least 5 days before any behavioral testing. All experiments were performed during the dark cycle phase (8 A.M. to 5 P.M.).

### Viral preparations

Biosensors, optogenetic and control plasmids packaged into viral particles were purchased from Addgene. For fiber photometry experiments we used pAAV.CAG.Flex.GCaMP6m.WPRE.SV40 (#100839-AAV5, 2.6 × 10$^{13}$ GC/ml), pGP.AAV.CAG.Flex.-jGCaMP7s.WPRE (#104495-AAVrg, 1.1 × 10$^{13}$ GC/ml), pAAV.hSyn.dLight1.2 (#111068-AAV5, 8.7 × 10$^{12}$ GC/ml). For tracing and optogenetic experiments we used pAAV.hSyn.mCherry (#114472-AAV2, 2.6 × 10$^{13}$ GC/ml, location marker), pAAV.hSyn.DIO.EGFP (#50457-AAV5, 1.3 × 10$^{13}$ GC/ml and -AAVrg, 1.4 × 10$^{13}$ GC/ml), pAAV.hSyn.DIO.mCherry (#50459-AAV5, 1.8 × 10$^{13}$ GC/ml and -AAVrg, 1.5 × 10$^{13}$ GC/ml), pENN.AAV.hSyn.Cre.WPRE.hGH (#105553-AAV1, 2.1 × 10$^{13}$ GC/ml), pAAV.Syn.Flex.rc[ChrimsonR-tdTomato] (#62723-AAV5, 8.5 × 10$^{12}$ GC/ml), pAAV.Ef1a.double floxed-hChR2(H134R).mCherry.WPRE.HGHpA (#20297-AAVrg 1.2 × 10$^{13}$ GC/ml), pAAV-hSyn-hChR2(H134R)-eYFP (#26973-AAV5 1.8 × 10$^{13}$ GC/ml). The viral stock pAAV.Ef1a.DIO.-eNpHR3.0.EYFP (#AV9115-rAAV2, 5.8 × 10$^{12}$ VM/ml) was obtained from UNC GTC Vector Core. Viral injections were performed between 4–6 weeks before experiments to allow adequate time for transgene expression.

### Stereotaxic surgeries

Surgeries were performed under aseptic conditions as previously described[15]. Briefly, mice (6–8 weeks old) were deeply anaesthetized via intraperitoneal (IP) injection of a 100 mg/kg ketamine (VEDCO) and 10 mg/kg xylazine (LLOYD) mixture. Ophthalmic ointment was applied to maintain eye lubrication. Following anesthesia, the surgical area was shaved and disinfected with iodine. Mice were then placed on a stereotaxic frame (Stoelting Co.) where bregma and lambda landmarks were used to level the skull along the coronal and sagittal planes. A 0.4-mm drill was used for craniotomies at the target Bregma coordinates. For viral injections, mice were microinjected at a controlled rate of 50 nl/min using a gas-tight 33 G 10-µl neurosyringe (1701RN; Hamilton) in a microsyringe pump (Stoelting Co). After injection, the needle remained unmoved for 10 min before a slow withdrawal. Injection coordinates were (in mm, Bregma anteroposterior (AP), mediolateral (ML), dorsoventral (DV) and angle): IPN (−3.4, −0.5, −4.86, 6°), VTA (−3.4, ± 0.35, −4.25, 0°), NAc (+1.4, −0.9, −4.2, 0°), LDTg (−5.1, ± 0.4, −3.1, 0°) and DR (−4.36, + 0.5, −3.04, 10°). Viral volumes for injections were 300 nl (IPN), 500-800 nl (VTA and NAc), 300 nl/site (LDTg), 500 nl (DR). For fiber photometry or optogenetic experiments, 3–5 weeks post-viral injection, an optic fiber implant (200-µm core diameter; 0.53 N.A., Doric Lenses) held in a magnetic aluminum

receptacle (Doric Lenses) was placed above the injection site, IPN (−3.4, −1.0, −4.82, 12°), VTA (−3.4, ±0.35, −4.2, 0°) or NAc (+1.4, −0.8, −4.1, 0°). For experiments that combined optogenetic photostimulation together with photometry recordings, a bilateral fiber implant was placed either above the VTA (−3.4, ±0.35, −4.2, 0°) or the LDTg terminal area (−5.1, ±0.4, −3.0, 0°) and the same mice received a fiber implant targeting the NAc (+1.4, −0.8, −4.1, 0°). All optic fiber implants were secured into the skull using adhesive (C&B Metabond cement, Parkell Inc.) followed by dental cement (Cerebond, PlasticsOne). After each surgery, mice received IP injections of 1 mg/kg ketoprofen analgesic (Zoetis) and placed on heating pads to be monitored post-surgery until sternal. Mice were allowed to recover in their home cages for 5 days before any behavioral testing. Injection sites and viral expression were confirmed for all animals by experimenters blinded to behavioral outcome as previously described[15]. Briefly, images from serial coronal sections encompassing the target brain areas of experimental animals were imaged and acquired using a fluorescent microscope (Zeiss, Carl Zeiss MicroImmagine, Inc., NY, USA) connected to computer-associated image analyzer software (Axiovision Rel., 4.6.1), as described below, and viral expression was visualized using the endogenous fluorescence of the virus. Animals showing no viral or off-target site viral expression or incorrect optic fiber placement (<10%) were excluded from analysis.

## Fiber photometry and data analysis

Florescent signals from biosensors were recorded with a Doric Instruments Fiber Photometry System as previously described[47]. Briefly, an LED driver was used to deliver excitation light from LEDs at 465 nm and at 405 nm (-30–60 µW output at fiber tip), which was used as an isosbestic wavelength for the indicator (Doric Instruments). The light was reflected into a 200 µm 0.53 N.A. optic fiber patch cord via the Dual Fluorescence Minicube (Doric Instruments). Emissions were detected with a femtowatt photoreceiver (Model 2151, Newport) and were amplified by transimpedance amplification to give an output voltage readout. Sampling (12 kHz) and lock-in demodulation of the fluorescence signals were controlled by Doric Neuroscience Studio software with a decimation factor of 50. A Doric behavior camera was connected to the Doric Neuroscience Studio software using USB 3.0 Vision interface to synchronize the photometry recordings with time-locked behavioral tracking systems. All mice were habituated to the patch cord plugged to the optic fiber implant for 10 min in their home cages prior to the start of the experiment. For novelty experiments, recordings began with the animal in the home cage for 1 min and then was placed by the experimenter to the center of the behavioral apparatus. Behavioral events were tallied from the videos in a blinded fashion and analysis was done using the time-locked photometry recording.

Fiber photometry data analysis was performed using custom-written Matlab and Python scripts. A lowpass filter (3 Hz) was applied to the demodulated fluorescence signals before the 405 nm channel was scaled to the 465 nm by applying a least mean squares linear regression. Scaled signals were used to calculate the $\Delta F/F_0$ where $\Delta F/F_0 = (465 \text{ nm signal} - \text{fitted } 405 \text{ nm signal})/\text{fitted } 405 \text{ nm signal}$. Z-scores were calculated using as baseline the average $\Delta F/F_0$ values from the −3.0 s prior to the onset of each behavioral exploratory event (considered as time zero, t = 0). Social investigations with an intertrial interval inferior to 2.5 s were excluded from analysis. The peak onset was established as the lowest absolute value closest to the initiation of a social investigation (t = 0). The peak latency was determined as the time difference between the peak onset and the maximum absolute value closest to t = 0. Area under the curve (AUC) of the z-score curves were calculated using the trapezoidal method in Python scripts. The mean z-score and mean AUC were estimated as average signal from −1s to +5 s upon initiation of a social investigation. For termination analysis, the mean z-score corresponded to the average signal from 0

to +2 s upon finalizing a social investigation. For the experiments that determined the effects of optogenetic photostimulation on NAc DA release, recordings were performed in mice remaining in their home cages. Photometry recordings were taken for 15 min with light photostimulation delivered during the 5-10 min period (Chrimson: 595 nm, 20 Hz, 12 ms pulse for 5 min) and DA peak activity was estimated using custom Matlab scripts.

## Behavioral assays

Animals were acclimated to the testing room for 30 min before any experimental assay, and all testing was performed under dim red-light conditions.

Social behavior experiments were performed in male mice that interacted with male juvenile conspecific, as previously reported[15]. Animals were tested in a plexiglass apparatus (42 × 64 × 30 cm) containing two plastic grid cylinders (25 cm × 10 cm diameter) located at opposite corners of a rectangular maze. Subject mice were first habituated to the apparatus and the empty cylinders for a 5-min period. Following habituation, a juvenile unfamiliar C57BL/6 J conspecific (4–7 weeks of age) was placed inside one of the two cylinders (counterbalanced), reducing social investigations led by the subject animals. The subject mouse was then positioned in the central zone and allowed to freely explore the social and non-social cylinders for 5 min. This testing phase was repeated 24 h and 48 h later, on day 2 and 3, using the same juvenile conspecific located in the same compartment. On day 4, a familiar conspecific (3 times encountered) was placed simultaneously with a novel, unfamiliar C57BL/6 J juvenile conspecific inside the opposite cylinder (counterbalanced) and exploratory behavior of the subject animal was measured for 5 min. The apparatus and cylinders were cleaned with Micro-90 solution (International Products Corporation) to eliminate olfactory traces after each session. All sessions were video recorded and/or synchronized to activity dynamics. Exploration of the social and non-social cylinders in videos of the trials were labeled frame by frame by experimenters blind to group conditions. Onset of each behavioral exploratory event (considered as t = 0) was defined whenever the subject mouse directed its nose towards the cylinders at a distance < 2 cm and initiated a social investigation. Sitting or resting next to the cylinder or objects was not considered exploration. All behavioral labeling was imported to Matlab for further analysis using custom-written scripts. For the choice experiments, the NP ratio was calculated as: (time interacting with the social cylinder with novel conspecific − time interacting with the social cylinder with familiar conspecific)/(total time interacting with cylinders) over the 5 min session.

For optogenetic experiments, optic fiber implants were connected to a patch cable (Doric Lenses) and a commutator (rotary joint; LEDFRJ-B_FC for blue light and LEDFRJ-A_FC for yellow light, Doric Lenses), by means of an FC/SMC adapter to allow unrestricted movement, in a manner identical to previous reports[15,48]. On day 1, mice were subjected to a 5-min habituation session to the maze, followed by a 5-min test session in which a juvenile C57BL/6 J unfamiliar conspecific was placed inside one of the two cylinders (counterbalanced). This test session was repeated 24 h later, with the same social stimulus located in the same cylinder. On day 3, a familiar conspecific (2 times encountered) was placed simultaneously with a second unfamiliar C57BL/6 J juvenile conspecific inside the opposite cylinder offering a choice to explore the two stimuli for 5 min. A high-power LED driver (DC2200, Thorlabs) was used to generate light pulses at intensity -5 mW at the fiber tip. Light photostimulation (Chrimson: 593 nm, 8 ms at 30 Hz for VTA DA neurons or 12 ms pulses at 20 Hz for the IPN$^{GAD2}$→LDTg circuit; NpHR3.0: 593 nm, constant light; ChR2: 473 nm, 20 Hz, 12 ms pulse) was delivered in time-locked mode by an experimenter blind to animals' conditions whenever the subject animal entered the cylinder zone. The optogenetic parameters used for DA VTA photostimulation were consistent with a previously published

protocol for DAergic firing. We chose to stimulate neurons with Chrimson instead of ChR2 because the stimulation wavelength has little overlap with eGFP-derived biosensor wavelengths. For all the optogenetic experiments, light was delivered in a closed-loop mode on the third session of NP. All sessions were video recorded from above (HDR-CX4440 camera, SONY).

### Electrophysiological recordings

Slice Preparation. We prepared coronal slices from fresh brain tissue of 8–9 week old ChAT$^{Cre}$ mice. Mice were deeply anesthetized with isoflurane prior to intracardiac perfusion with an ice-cold N-methyl-D-glucamine-based solution (see below). Brains were rapidly removed and transferred into a cold (~ + 0.5 °C) oxygenated (95% $O_2$ and 5% $CO_2$) cutting solution of the following composition (in mM): 95 N-methyl D-glucamine (NMDG), 2 thiourea, 5 Na$^+$-ascorbate, 3 Na$^+$-pyruvate. Slices (200 μm) were cut with a Vibroslicer (VT1200, Leica MicroInstruments; Germany) and immediately transferred in an incubation chamber where they were left to recuperate in the NMDG-based solution for 20 min at 30 °C before being moved into a chamber containing artificial cerebrospinal fluid (ACSF; in mM): 126 NaCl, 2.5 KCl, NaH$_2$PO$_4$.H$_2$O, 1 MgCl$_2$, 2 CaCl$_2$, 26 NaHCO$_3$, 10 D-Glucose, at ambient temperature. Slices were left in this chamber for at least one hour before being placed in a recording chamber and perfused with ACSF at a constant rate of 2–3 ml/min at room temperature (~21 °C).

Ex vivo Optogenetic Stimulation. Whole-cell patch clamp recordings were performed in the presence of 30 μM NBQX and 25 μM D-AP5 AMPA and NMDA receptor antagonists, respectively. Briefly, borosilicate glass electrodes (1.5 mm OD, 5–8 MΩ resistance) were filled with an internal solution containing (mM): 120 Cesium-methanesulfonate; 20 KCl; 10 HEPES; 2 ATP, 1 GTP, and 12 phosphocreatine. We identified mCherry labeled LDTg neurons receiving inputs from the IPN using a 585 nm light source passed through a 645/75 emission filter and a 560/55 excitation filter (Chroma, Bellows Falls, VT) at 60X magnification (CooLED, NY, United States). Following seal rupture, the fluorescent cell was recorded in voltage clamp recording mode. Recordings were performed with holding potentials (mV) of −90, −80, −70, −50, −40, and −30. Simultaneously, Inhibitory Post-Synaptic Currents (IPSCs) were optogenetically evoked by flashing 1 ms-long 470 nm blue light through the light path of the microscope 60X objective using high powered LED (pE-100 470 nm CooLED, NY, United States). To determine whether the optically evoked post-synaptic currents at each holding potential were mediated by GABA, ionotropic GABA$_A$ receptors were inhibited with 50 μM picrotoxin (PTX) and evoked currents were recorded as above. PTX was perfused through the recording chamber for 5 min prior to IPSC recordings. Voltage traces in whole-cell patch-clamp were acquired with an EPC10 amplifier and the PatchMaster v2x90 software (HEKA Electronik; Germany). All traces were subsequently analyzed offline with FitMaster 2.15 (HEKA Electronik; Germany).

### Immunostaining and microscopy

Immunohistochemistry and microscopy were performed as described previously[15]. In brief, mice received a euthanizing dose of sodium pentobarbital (200 mg/kg) before they were transcardially perfused with ice-cold 0.1 M phosphate buffer saline (PBS, pH 7.4) followed by 10 ml of cold 4% (W/V) paraformaldehyde (PFA) in 0.1 M PBS. Brains were post-fixed for 4 h in 4% PFA and transferred to a 30 % sucrose PBS solution for 48 h. Coronal sections (25 μm) were obtained using a freezing microtome (HM430; Thermo Fisher Scientific, MA, USA). For immunohistochemical experiments, brain sections were permeabilized with 0.5% Triton X-100 (Sigma) in 0.1 M PBS for 10 min, blocked with 5% donkey serum (DS, Sigma) in 0.1 M PBS for 30 min and then incubated overnight (o.n.) with the corresponding primary antibodies in 0.1 M PBS 3% DS at 4 °C. Primary antibodies used: goat anti-ChAT

1:400 (Millipore, AB144P, RRID:AB_2079751), goat anti-VIAAT (1 mg/ml, Nittobo-nmd, VGAT-Go-Af620, RRID:AB_2571623), guinea pig anti-synaptophysin (1 mg/ml, Nittobo-nmd, Syn-GP-Af300, RRID: AB_2571843), mouse anti-tyrosine hydroxylase 1:700 (Millipore, MAB318, RRID:AB_2201528) and rabbit anti-Tph2 1:500 (Abcam, ab111828, RRID:AB_10862137). Slices were subsequently washed in 0.1 M PBS and incubated in secondary antibodies for 2 h (1:800; Life Technologies; donkey anti-goat 594 (A11058, RRID:AB_2534105), goat anti-guinea pig 594 (A11076, RRID:AB_141930), donkey anti-mouse 594 (R37115, RRID:AB_141633) and donkey anti-rabbit 594 (R37119, RRID:AB_141637)). After washes in 0.1 M PBS, nuclei were counterstained with DAPI, sections were mounted, air-dried and coverslipped with Mowiol (Sigma). All slices were imaged using a fluorescent microscope (Zeiss, Carl Zeiss MicroImmagine, Inc., NY, USA) connected to computer-associated image analyzer software (Axiovision Rel., 4.6.1).

### Statistical analysis and reproducibility

Data were analyzed by means of two-tailed unpaired $t$ test, one-way or two-way ANOVAs with/without repeated-measures (RM), as indicated. Šidák's test was used as a *post hoc* for multiple comparisons. Two-tailed Pearson r was used for correlation analysis. Comparisons of z-scores or AUC photometry signals were made using the calculated average for each animal. Mean z-score or AUC values were estimated as the average signal per event from −1 to 5 s from the onset of social investigation. Each data set was tested for normal distribution prior to analysis and presented as mean ± standard error of the mean (SEM). All behavioral experiments were conducted in at least two independent cohorts of animals. Imaging analyses included representative pictographs of at least three animals included in the same experiment. All statistical analyses were performed in GraphPad Prism 9.3.0. Software (Graphpad Software Inc.) and statistical significance was established at p < 0.05.

### Reporting summary

Further information on research design is available in the Nature Portfolio Reporting Summary linked to this article.

## Data availability

The data generated in this study and used to produce the figures are provided in the Source Data file. The raw fiber photometry data are available under restricted access as they are still in use at the time of publication. Data will be made available on request to the corresponding author. Source data are provided with this paper.

## Code availability

Custom codes used in this manuscript are publicly accessible on Harvard Dataverse using the following link: https://doi.org/10.7910/DVN/BXCYZS.

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

## Acknowledgements

We thank Anthony Sacino for technical support. This work was supported by the National Institute on Drug Abuse award numbers DA041482 (A.R.T.), DA047678 (A.R.T.) and a Brain and Behavior Research Foundation Young Investigator Award (S.M.).

## Author contributions

S.M., R.Z., G.E.M., and A.R.T. contributed to the study design. S.M., T.G.F., R.Z., T.L., P.G., and M.B. performed the experiments and analyzed the data. S.M., R.Z., G.E.M., and A.R.T. prepared the manuscript. All authors contributed to editing and reviewing the manuscript.

## Competing interests

All authors report no competing financial interests or potential conflicts of interest. The content is solely the responsibility of the authors and does not necessarily represent the official views of the National Institutes of Health.
