## [Peer Review File · Nature Communications]

Dopamine control of social novelty preference is constrained by an interpeduncular-tegmentum circuitREVIEWER COMMENTS

Reviewer #1 (Remarks to the Author):

The study investigated the mechanisms underlying social novelty preference in mice. Using calcium imaging, fiber photometry and optogenetics in the VTA, NAC, IPN and LDT, the authors found that VTA DA to NAC projections may contribute to the rewarding effect of social novelty, while IPN GABAergic neurons projecting to the LDT may contribute to the termination of activity with a familiar social stimuli. The questions asked are very important and relevant to a number of behaviors and mental disorders. The experimental design is outstanding, and the results are mostly very clear (see concern below). The manuscript is well rewritten and the conclusions appropriate. Overall, this is an outstanding manuscript.

Concerns

1. The results of Fig 3 are difficult to interpret because there is no DA release in the control during social novelty. It is unclear why the control animals in Fig. 3N do not show much increase in DA compared to the control animals in Fig 2C.
2. Related to the point above, it appears that the phenomenon observed (DA/social novelty) is not very strong as sometimes the effect is not observed in most animals (e.g. see fig. 6h)
3. Fig 4 D should show the signal before the termination (-2 to 0s).
4. The lack of statistical testing when comparing the timecourse of all photometry data makes it difficult to interpret the results.
5. The results, abstract, and title should state that these results have been obtained only in males.
6. The figures are sometimes difficult to understand with missing labels. (E.g. "Z score", Z score of what? The dependent variable that is z-scored should be indicated) Same thing for the heatmap that do not show what measure is represented in the heatmap, also, some axis have no ticks/labels making it difficult to evaluate the results.

Reviewer #2 (Remarks to the Author):

The authors have explored the neural circuitry contributing to the behavioral responses to novel vs familiar social interactions in mice. They employed an impressive array of behavioral analyses combined with genetic tools to monitor activity in specific cell types within the mesoaccumbens dopamine system and in IPN LDTg projections. They confirm the necessity of specific projections using optogenetic excitation/inhibition. Confirmation of these manipulations and the impact on dopamine release was provided by in vivo monitoring of extracellular dopamine in the nucleus accumbens. Overall, this study brings reveals and important insight into the behavioral consequences of IPN activity, with some of the strongest evidence for interactions between that nucleus and midbrain dopamine neurons in the VTA. The observation that IPN activity governs the reduction in time spent in familiar social interactions provides an intriguing and important extension of our current understanding of the role of that nucleus, which has been implicated most prominently in aversive experience and anxiety. The manuscript is very well-written and logically organized, and the content will be of interest to specialists in the field and to the broad readership of the journal.

My only concern with the study is related to the evidence implicating LDTg-projecting IPN neurons specifically. The authors provide support for the selective expression of optogenetic probes to the IPN-LDTg projection neurons with viral injections that are restricted to the LDTg. IPN axonal projections to Raphe nuclei are located very close to if not overlapping with the LDTg, which raises the possibility that the IPN outputs that the authors are labeling with their LDTg injections are modulating other centers. As Dorsal Raphe outputs are known to affect VTA dopamine neuron activity, it is important that the authors demonstrate that their viral injections are not also labeling IPN-DRN projections.

Reviewer #3 (Remarks to the Author):

Molas and colleagues present studies that explored a role of mesolimbic dopamine (DA) in social interactions, focusing on social novelty and familiarity and how these factors are encoded by ventral tegmental area (VTA) DA neurons to express social novelty preference (NP). In addition, they describe a circuit of GABAergic projections from the interpeduncular nucleus (IPN) to the latero-dorsal tegmentum (LDTg) that regulates NP by decreasing DA neurotransmission. The present manuscript is an extension of their previous 2017 study (Molas et al.) in which they described how IPN GABAergic activity bidirectionally modulates social NP and how VTA DA inputs to IPN control social NP. This manuscript includes results that are of interest to the field, but conceptually, provide a modest advance given that the core ideas have been already described and explored in previous work published by the authors and others. The authors even used very similar phrases in the two studies, for example: discussion from the 2017 article: "Thus, when given a choice between novel and familiar stimuli, the IPN acts as a brake that is necessary and sufficient for reduced exploration of a familiar but not novel signal to control NP." And line 206 from the present manuscript: "Together, these results indicate that increased IPN GAD2-LDTg circuit activity observed with the termination of familiar social investigations acts as a brake on lengthy novel exploratory events to control social NP responses".

Major comments:

- Given the previous publications by the authors and others, the novel contribution of this manuscript resides in the potential role in social novelty preference by the IP inhibitory regulation on LDTg neurons regulating VTA dopamine neurons. However, the authors did not provide results supporting the specific manipulation and recording within the LDTg. Similarly, the authors did not provide direct evidence indicating that specific inputs from IP to LDTg regulate the firing of dopamine neurons.
 - In the introduction section, some concepts must be revised. For example, social behaviors are not "experience-based motivational drives". Social behavior is an innate behavior in which approach or avoidance to a conspecific may be experience-based. Also, a behavior is not a drive: internal or external drives will elicit or disrupt behavior. In addition, the authors explicitly say that "behaviors associated with social experiences reflect rewarding events...", which is not always the case. Social encounters resulting in fights for partners or resources are not rewarding events.
 - In the results section (lines 161-164), authors present data from their previous article (Molas et al., 2017) that have been reanalyzed. They must provide a better framework introducing the experiment aim and how it was conducted. Also in line 164, replace "behaviors" to "events".
 - The authors must present histological diagrams showing the location of all the optic fibers (photometry and optogenetic fibers) from their experimental mice.
 - Only male mice were used in this study. Authors should repeat some of the experiments to show that their results are also observed (or not) in female mice.
 - The authors concluded that interpeduncular nucleus (IPN) GABAergic neurons that project to the lateral dorsal tegmentum (LDTg) were inhibited by social novelty but activated during terminations with familiar social stimuli. However, the authors did not provide results (images and diagrams) demonstrating that the IPN GABAergic inputs were confined to the LDTg.
- The authors suggested that IPN-GABAergic neurons inhibited lateral dorsal tegmentum (LDTg) cholinergic neurons that in turn control VTA dopamine neurons. However, the authors did not provide results indicating that IPN-GABAergic neurons establish monosynaptic synapses on cholinergic neurons that synapse on VTA dopamine neurons. Thus, the authors need to provide electrophysiological evidence supporting the claim of inhibitory regulation on LDTg cholinergic by IPN-GABAergic neurons.
- Given that surrounding areas to LDTg receive inputs from IP and project to VTA, it is necessary that the authors provide evidence indicating that viral injections or recordings were indeed within the LDTg, for instance, by providing data on the rostro-caudal distribution of viral expression in relation to immunodetection of ChAT.
 - In the discussion section (lines 270-273), authors imply that novelty or familiarity of given stimuli are correlated with the aversiveness of the stimuli, and this is not correct. This sentence should be

rephrased. In addition, the last part of the discussion is focused in the "satiety of social interactions", but the authors never directly tested satiety of social interactions. They inferred social satiety from their experiments by equating familiarity with satiety, which are two different concepts. The authors must tone-down the discussion related to satiety of social interactions or do an experiment addressing specifically this topic.

Minor comments:

-The authors stated that "LDTg sends strong cholinergic excitatory inputs to the VTA DA systems, and these projections have been implicated in DAergic neuron firing rates and reward-related behavior". However, it is well known that LDTg GABAergic and LDTg glutamatergic neurons innervate the VTA.

-The number of mice excluded from the study because no viral or off-target site viral expression or incorrect optic fiber placement should be specified.

-Authors should specify if they imaged sections showing endogenous fluorescence for the viral vectors used or if they enhanced the endogenous fluorescence of the vectors by use of antibodies, in which case, they must describe the antibodies used.

-For the symbols depicting the significance of the comparisons, the authors must specify which symbols represent which comparisons since this is not clear in the figures or figure legends.

-Figure legend for supplementary figure 1e, 1f and 1g: replace "m" by "d".

-Consider using transparency to show the overlapping traces in all the figures presenting z-scores (1d, S1d, S1h, 2c, S2d, S2h, 3e, 3n, 4b, 4d, 5b, 5d, 6h). Currently some of the traces cover the others, making it difficult to draw conclusions on these results.

-Figures 3j, 3l, S3b, S3c, 5h, 5k, S4g, 6b, 6e must be analyzed by ANOVA or must be presented in separated panels to allow for a t test analysis.

-Figures 1d, S1d, S1h, 2c, S2d, S2h, 3n, 4b, 5b and 6h should show a red line at time 0, as described in the figure legends or authors must specify that a white line is used in the heatmaps.

-Figures 1b, 4a, 5a should show the ending tip of the optic fiber, not the length of the fiber.

-Figure 1a, it is not clear what the triangle above habituation means.

-In figure 2a, it is not clear what the diagram to the right of the picture is indicating since the light blue rectangle is located to the left of the anterior commissure and the spread of the viral injection is below and to the right of the anterior commissure in the image shown. Please, clarify.

-In figure 3e, the VTA diagram is not described, need to include the description in the figure legend or remove the diagram.

-Figure 3m should be located after figure 3n because it depicts the values and analyses of the results shown in figure 3n.

-Figures 4d and 5d should show the values of the z-scores before timepoint 0 (when the mice are still interacting with the conspecific) to see how the IPN neuros or the IPN-LDTg circuit are acting before the termination of the social encounter.

-High magnification images showing some of the neurons indicated by white arrowheads in figures 4g, 4i and S3h should be provided. Currently, it is very difficult to see the neurons indicated in these figures.

-Figure 5a should be divided: the top 2 panels as figure 5a and bottom 2 panels as figure 5b. The rest of the panels should be renamed accordingly.

-Figure S4a should be moved to a different figure (5) since it doesn't belong with the group of results showed in figure S4.

-Use pre-photostimulation instead of basal in figures S4b and S4c.

-Figures 6f and 6g should be located after figure 6h because they depict the values and analyses of the results shown in figure 6h.

We thank the reviewers for their thoughtful comments regarding our manuscript. In response, we have performed several new experiments and data analysis, as well as added new discussion points. We address Reviewer comments on a point-by-point basis below. Textual changes in the manuscript are highlighted in yellow. Also note that figure numbers have changed since we added a new Figure 6 and also have included four new Supplementary Figures (S2, S4, S7, and S8).

REVIEWER COMMENTS

Reviewer #1 (Remarks to the Author):

The study investigated the mechanisms underlying social novelty preference in mice. Using calcium imaging, fiber photometry and optogenetics in the VTA, NAC, IPN and LDT, the authors found that VTA DA to NAC projections may contribute to the rewarding effect of social novelty, while IPN GABAergic neurons projecting to the LDT may contribute to the termination of activity with a familiar social stimuli. The questions asked are very important and relevant to a number of behaviors and mental disorders. The experimental design is outstanding, and the results are mostly very clear (see concern below). The manuscript is well rewritten and the conclusions appropriate. Overall, this is an outstanding manuscript.

Concerns

1. The results of Fig 3 are difficult to interpret because there is no DA release in the control during social novelty. It is unclear why the control animals in Fig. 3N do not show much increase in DA compared to the control animals in Fig 2C.

The data presented in Fig 2C represent a first social encounter with a single conspecific; whereas, data from Fig 3N represent dopamine signal during social novelty preference where the test animal has a choice between a novel and familiar conspecific and has had two previous social trials prior to the preference test. In our experience, accumbens dopamine signal to the initial social novelty encounter is always greater than subsequent social interactions even when we include a new social stimulus in a new spatial location within the test chamber. We have added this consideration of the effect of social context on subsequent DA and neural responses in the discussion section.

2. Related to the point above, it appears that the phenomenon observed (DA/social novelty) is not very strong as sometimes the effect is not observed in most animals (e.g. see fig. 6h)

As discussed above, DA signals are more variable during the novelty preference test as compared to initial single social encounters. We note that in Fig 6h (now Fig 7h) the majority of animals display a positive mean Z-score during social novelty exploration in the novelty preference test.

3. Fig 4 D should show the signal before the termination (-2 to 0s).

We have now added -1 s prior to termination for both IPN GAD2 neurons (Fig. 4D) and IPN→LDTg neurons (Fig. 5D) as recommended.

4. The lack of statistical testing when comparing the timecourse of all photometry data makes it difficult to interpret the results.

In consideration of the Reviewers comment, because most of our photometry data analysis compares different conditions and multiple types of stimuli (i.e. novelty vs. familiarity, first

encounter vs. subsequent encounter, opsin photostimulation vs. control, etc.) rendering comparison of time course across experiments difficult, we believe that comparison of mean z-scores within groups/conditions may accurately reflect differences in activity patterns across experiments.

5. The results, abstract, and title should state that these results have been obtained only in males.

Based on this concern and that of reviewer 3, we have repeated the main photometry results including GCaMP responses in DAergic neurons and DA signals in NAc during novel/familiar social encounter and novelty preference in females. While the overall results of these experiments is similar in male and female mice, our data indicate that the engagement of DA with social novelty is significantly lower in females compared to males so we present the female data separately as supplementary (Figures S2, S4, and S5).

6. The figures are sometimes difficult to understand with missing labels. (E.g. "Z score", Z score of what? The dependent variable that is z-scored should be indicated) Same thing for the heatmap that do not show what measure is represented in the heatmap, also, some axis have no ticks/labels making it difficult to evaluate the results.

We thank the reviewer for pointing this out and we have changed all the axes and now indicate Mean Z-score or Mean AUC, corresponding to -1 to +5 s of the signal, and specified this in the methods section.

Reviewer #2 (Remarks to the Author):

The authors have explored the neural circuitry contributing to the behavioral responses to novel vs familiar social interactions in mice. They employed an impressive array of behavioral analyses combined with genetic tools to monitor activity in specific cell types within the mesoaccumbens dopamine system and in IPN LDTg projections. They confirm the necessity of specific projections using optogenetic excitation/inhibition. Confirmation of these manipulations and the impact on dopamine release was provided by in vivo monitoring of extracellular dopamine in the nucleus accumbens. Overall, this study brings reveals and important insight into the behavioral consequences of IPN activity, with some of the strongest evidence for interactions between that nucleus and midbrain dopamine neurons in the VTA. The observation that IPN activity governs the reduction in time spent in familiar social interactions provides an intriguing and important extension of our current understanding of the role of that nucleus, which has been implicated most prominently in aversive experience and anxiety. The manuscript is very well-written and logically organized, and the content will be of interest to specialists in the field and to the broad readership of the journal.

My only concern with the study is related to the evidence implicating LDTg-projecting IPN neurons specifically. The authors provide support for the selective expression of optogenetic probes to the IPN-LDTg projection neurons with viral injections that are restricted to the LDTg. IPN axonal projections to Raphe nuclei are located very close to if not overlapping with the LDTg, which raises the possibility that the IPN outputs that the authors are labeling with their LDTg injections are modulating other centers. As Dorsal Raphe outputs are known to affect VTA dopamine neuron activity, it is important that the authors demonstrate that their viral injections are not also labeling IPN-DRN projections.

We thank the reviewer for their suggestion and now present images and localizations of injections demonstrating that retrograde viral injections of NpHR were restricted to the LDTg area, using

ChAT immunostaining to delineate this area (Fig. S7a-b). In addition, to alleviate the reviewer's concern, we injected retrograde Cre-dependent NpHR directly in the dorsal raphe (DR) and placed a fiber in the IPN to silence the IPN→DR GAD2+ circuit. Optogenetic manipulations of this circuit did not affect social novelty preference and we present these important control data in Fig. S7c-g.

Reviewer #3 (Remarks to the Author):

Molas and colleagues present studies that explored a role of mesolimbic dopamine (DA) in social interactions, focusing on social novelty and familiarity and how these factors are encoded by ventral tegmental area (VTA) DA neurons to express social novelty preference (NP). In addition, they describe a circuit of GABAergic projections from the interpeduncular nucleus (IPN) to the latero-dorsal tegmentum (LDTg) that regulates NP by decreasing DA neurotransmission. The present manuscript is an extension of their previous 2017 study (Molas et al.) in which they described how IPN GABAergic activity bidirectionally modulates social NP and how VTA DA inputs to IPN control social NP. This manuscript includes results that are of interest to the field, but conceptually, provide a modest advance given that the core ideas have been already described and explored in previous work published by the authors and others. The authors even used very similar phrases in the two studies, for example: discussion from the 2017 article: "Thus, when given a choice between novel and familiar stimuli, the IPN acts as a brake that is necessary and sufficient for reduced exploration of a familiar but not novel signal to control NP." And line 206 from the present manuscript: "Together, these results indicate that increased IPN GAD2-LDTg circuit activity observed with the termination of familiar social investigations acts as a brake on lengthy novel exploratory events to control social NP responses".

Major comments:

-Given the previous publications by the authors and others, the novel contribution of this manuscript resides in the potential role in social novelty preference by the IP inhibitory regulation on LDTg neurons regulating VTA dopamine neurons. However, the authors did not provide results supporting the specific manipulation and recording within the LDTg. Similarly, the authors did not provide direct evidence indicating that specific inputs from IP to LDTg regulate the firing of dopamine neurons.

We thank the reviewer for their comment, we believe that our manuscript will be an important contribution to the literature as it specifies that dopamine directly controls bout length of an initial novel social encounter. In addition, it is one of the initial studies investigating IPN neuronal activity dynamics synchronized with animal behavior in general and social behavior specifically, so all data regarding contribution of in vivo IPN neuronal activity to social novelty and familiarity signaling is new. In addition, it is the first to show that IPN activity via innervation of the LDTg can control DA signaling in the nucleus accumbens. To better connect the IPN→LDTg circuit to the VTA, we have included immunostaining and electrophysiological recordings which demonstrate that the IPN-LDTg circuit is directly connected to the VTA DA system. This is done using a combination of retrograde viral injections in the VTA to fluorescently tag LDTg→VTA neurons in a Cre-dependent manner in C57Bl/6J mice and expressing Cre through transsynaptic delivery via an AAV1-Cre injection in the IPN resulting in fluorescent labeling of VTA projecting LDTg neuron that receive input from the IPN. We complement this experiment with slice electrophysiology demonstrating IPN GABAergic inputs monosynaptically innervate LDTg cholinergic neuron that project to the VTA. These data are shown in new Fig. 6 and Fig. S8.

-In the introduction section, some concepts must be revised. For example, social behaviors are not “experience-based motivational drives”. Social behavior is an innate behavior in which approach or avoidance to a conspecific may be experience-based. Also, a behavior is not a drive: internal or external drives will elicit or disrupt behavior. In addition, the authors explicitly say that “behaviors associated with social experiences reflect rewarding events...”, which is not always the case. Social encounters resulting in fights for partners or resources are not rewarding events.

We thank the reviewer for pointing out our error and have deleted reference to “experience-based motivational drives” and we have also specified in the introduction that “behaviors associated with positive social experiences reflect rewarding events.”

-In the results section (lines 161-164), authors present data from their previous article (Molas et al., 2017) that have been reanalyzed. They must provide a better framework introducing the experiment aim and how it was conducted. Also in line 164, replace “behaviors” to “events”.

We have now better phrased this section and replaced “behaviors” with “events” in line 164.

-The authors must present histological diagrams showing the location of all the optic fibers (photometry and optogenetic fibers) from their experimental mice.

Within the supplementary figures we now include diagrams indicating fiber placement for all animals used in this study (Fig. S1a, S2a, S3a, S5c, S6a and i, S7 and g, and S9a and g).

-Only male mice were used in this study. Authors should repeat some of the experiments to show that their results are also observed (or not) in female mice.

As indicated in response to Reviewer 1, we have repeated the main photometry results including GCaMP responses in DAergic neurons and DA signals in NAc during novel/familiar social encounter and novelty preference in females. While the overall results of these experiments is similar in male and female mice, our data indicate that the engagement of DA with social novelty is significantly lower in females compared to males so we present the female data separately as supplementary (Figures S2, S4, and S5).

- The authors concluded that interpeduncular nucleus (IPN) GABAergic neurons that project to the lateral dorsal tegmentum (LDTg) were inhibited by social novelty but activated during terminations with familiar social stimuli. However, the authors did not provide results (images and diagrams) demonstrating that the IPN GABAergic inputs were confined to the LDTg.

We now present images showing the viral injections were restricted in the LDTg area, using ChAT immunostaining to delineate this area. Also, in a control experiment, we injected retrograde Cre-dependent NpHR in the DR and placed a fiber in the IPN to silence the IPN→DR GAD2+ circuit. Optogenetic manipulations of this circuit did not affect social novelty preference, demonstrating specificity of our IPN→LDTg manipulations on novelty preference (See Fig. S7 and response to Reviewer 2).

-The authors suggested that IPN-GABAergic neurons inhibited lateral dorsal tegmentum (LDTg) cholinergic neurons that in turn control VTA dopamine neurons. However, the authors did not provide results indicating that IPN-GABAergic neurons establish monosynaptic synapses on cholinergic neurons that synapse on VTA dopamine neurons. Thus, the authors need to provide

electrophysiological evidence supporting the claim of inhibitory regulation on LDTg cholinergic by IPN-GABAergic neurons.

We thank the Reviewer for this suggestion. We have designed a viral strategy and performed electrophysiological recordings in acute mouse brain slices demonstrating monosynaptic connections between the IPN and LDTg→VTA circuit (New Fig. 6 and S8). Our new results provide first time evidence that the IPN establishes monosynaptic inhibitory connections with LDTg ChAT+ neurons that innervate the VTA. Thus, the IPN provides direct inhibition of LDTg→VTA ChAT+ neurons contributing to modulation of mesolimbic DA systems.

-Given that surrounding areas to LDTg receive inputs from IP and project to VTA, it is necessary that the authors provide evidence indicating that viral injections or recordings were indeed within the LDTg, for instance, by providing data on the rostro-caudal distribution of viral expression in relation to immunodetection of ChAT.

We now present images showing the viral injections were restricted in the LDTg area, using ChAT immunostaining to delineate this area (See Fig. S7a-b and response to Reviewer 2).

-In the discussion section (lines 270-273), authors imply that novelty or familiarity of given stimuli are correlated with the aversiveness of the stimuli, and this is not correct. This sentence should be rephrased. In addition, the last part of the discussion is focused in the “satiety of social interactions”, but the authors never directly tested satiety of social interactions. They inferred social satiety from their experiments by equating familiarity with satiety, which are two different concepts. The authors must tone-down the discussion related to satiety of social interactions or do an experiment addressing specifically this topic.

We have rephrased the sentence in line 270-273 and removed the word satiety from the discussion as recommended.

Minor comments:

-The authors stated that “LDTg sends strong cholinergic excitatory inputs to the VTA DA systems, and these projections have been implicated in DAergic neuron firing rates and reward-related behavior”. However, it is well known that LDTg GABAergic and LDTg glutamatergic neurons innervate the VTA.

We have included in the manuscript that LDTg GABAergic and glutamatergic neurons also innervate the VTA to control motivational behaviors and cite the relevant literature.

-The number of mice excluded from the study because no viral or off-target site viral expression or incorrect optic fiber placement should be specified.

We have included this statement in the methods: “Animals showing no viral or off-target site viral expression or incorrect optic fiber placement (< 10%) were excluded from analysis.”

-Authors should specify if they imaged sections showing endogenous fluorescence for the viral vectors used or if they enhanced the endogenous fluorescence of the vectors by use of antibodies, in which case, they must describe the antibodies used.

All the images of viral expression correspond to the endogenous fluorescence, this is now added to the methods section.

-For the symbols depicting the significance of the comparisons, the authors must specify which symbols represent which comparisons since this is not clear in the figures or figure legends.

All statistical analysis and symbols are described in all the figure legends.

-Figure legend for supplementary figure 1e, 1f and 1g: replace “m” by “d”.

This is now updated.

-Consider using transparency to show the overlapping traces in all the figures presenting z-scores (1d, S1d, S1h, 2c, S2d, S2h, 3e, 3n, 4b, 4d, 5b, 5d, 6h). Currently some of the traces cover the others, making it difficult to draw conclusions on these results.

We have used transparency for the error bar shadows.

-Figures 3j, 3l, S3b, S3c, 5h, 5k, S4g, 6b, 6e must be analyzed by ANOVA or must be presented in separated panels to allow for a t test analysis.

All figures indicated are now presented in separate panels.

-Figures 1d, S1d, S1h, 2c, S2d, S2h, 3n, 4b, 5b and 6h should show a red line at time 0, as described in the figure legends or authors must specify that a white line is used in the heatmaps.

We have changed the color of the white line in all the heatmaps to red color.

-Figures 1b, 4a, 5a should show the ending tip of the optic fiber, not the length of the fiber.

We always present in all our images the end of the optic fiber tip.

-Figure 1a, it is not clear what the triangle above habituation means.

We now indicate that the triangle means the start of the experiment.

-In figure 2a, it is not clear what the diagram to the right of the picture is indicating since the light blue rectangle is located to the left of the anterior commissure and the spread of the viral injection is below and to the right of the anterior commissure in the image shown. Please, clarify.

We now provide new diagrams in the supplementary information showing the viral injections and fiber locations for all the animals used in the study.

-In figure 3e, the VTA diagram is not described, need to include the description in the figure legend or remove the diagram.

We now include the description of the diagram in the figure legend.

-Figure 3m should be located after figure 3n because it depicts the values and analyses of the results shown in figure 3n.

We have updated the figure with a new order.

-Figures 4d and 5d should show the values of the z-scores before timepoint 0 (when the mice are still interacting with the conspecific) to see how the IPN neuros or the IPN-LDTg circuit are acting before the termination of the social encounter.

We have now added -1 s prior to termination for both IPN GAD2 neurons and IPN→LDTg circuit.

-High magnification images showing some of the neurons indicated by white arrowheads in figures 4g, 4i and S3h should be provided. Currently, it is very difficult to see the neurons indicated in these figures.

We have added high magnification images for all these figures.

-Figure 5a should be divided: the top 2 panels as figure 5a and bottom 2 panels as figure 5b. The rest of the panels should be renamed accordingly.

The figure has been updated.

-Figure S4a should be moved to a different figure (5) since it doesn't belong with the group of results showed in figure S4.

This figure has been moved next to the figures corresponding to the same experiment.

-Use pre-photostimulation instead of basal in figures S4b and S4c.

We have changed the word "basal" to "pre-photostimulation" in all the figures.

-Figures 6f and 6g should be located after figure 6h because they depict the values and analyses of the results shown in figure 6h.

We have updated the figure with a new order.

REVIEWERS' COMMENTS

Reviewer #1 (Remarks to the Author):

the authors have addressed my concerns

Reviewer #2 (Remarks to the Author):

The authors have addressed my concerns and completed additional experiments to address those of the other reviewers. This is a strong publication that will be of interest to the readership of the journal.

Reviewer #2 (Remarks on code availability):

I looked at the code, and found several routines that would accomplish the goal of the programs. I did not run the code.

Reviewer #3 (Remarks to the Author):

In this revised version of the manuscript, the authors addressed all my concerns raised previously.

I